# Planar Printed Structures Based on Matryoshka Geometries: A Review

**DOI:** 10.3390/mi15040469

**Published:** 2024-03-29

**Authors:** Alfredo Gomes Neto, Jefferson Costa e Silva, Joabson Nogueira de Carvalho, Custódio Peixeiro

**Affiliations:** 1Group of Telecommunications and Applied Electromagnetism (GTEMA), Instituto Federal da Paraíba, João Pessoa 58015-435, Brazil; alfredogomes@ifpb.edu.br (A.G.N.); jefferson@ifpb.edu.br (J.C.e.S.); joabson@ifpb.edu.br (J.N.d.C.); 2Instituto de Telecomunicações, Instituto Superior Técnico, University of Lisbon, 1049-001 Lisbon, Portugal

**Keywords:** printed circuits, microstrip, ring resonators, Matryoshka sets, frequency selective surfaces, filters, antennas, sensors

## Abstract

A review on planar printed structures that are based on Matryoshka-like geometries is presented. These structures use the well-known principle of Matryoshka dolls that are successively nested inside each other. The well-known advantages of the planar printed technology and of the meandered nested Matryoshka geometries are combined to generate miniaturized, multi-resonance, and/or wideband configurations. Both metal and complementary slot structures are considered. Closed and open configurations were analyzed. The working principles were explored in order to obtain physical insight into their behavior. Low-cost and single-layer applications as frequency-selective surfaces, filters, antennas, and sensors, in the microwave frequency region, were reviewed. Potential future research perspectives and new applications are then discussed.

## 1. Introduction

The name Matryoshka comes from the well-known Russian dolls, shown in Figure 1, that are successively nested inside each other. It has been used to refer to nested sets in many areas of electrical and electronics engineering, such as electronics packaging [1], implantable medical devices [2], biomedical imaging [3], computer network security [4], silicon compounds [5], software protection [6], Internet-of-Things [7], image retrieval [8] and reconstruction [9], cancer gene analysis [10], cellular biophysics [11], cloud computing [12], 5G network slicing [13], acoustic wave resonators [14], pattern recognition [15], and astronomy [16]. It has also been used associated with planar printed configurations. This combination of printed circuits with Matryoshka-like geometries benefits from the well-known advantages of printed planar technology (low profile, lightweight, compactness, low cost, easy fabrication and integration of electronic components, wide range of characteristic impedances) and the multiband (or wideband) behavior and miniaturization associated with Matryoshka configurations. The Matryoshka-like scheme was used for the first time in planar printed circuit structures at the Group of Telecommunications and Applied Electromagnetism (GTEMA) from the Instituto Federal da Paraíba in João Pessoa, Brazil. A multi-resonant frequency selective surface (FSS) was proposed in 2014 [17,18]. Since then, the work in these planar printed structures based on Matryoshka geometries has progressed steadily at GTEMA, as reported in many MSc theses [17,19,20,21,22,23,24,25,26,27,28,29,30,31] and associated publications [32,33,34,35,36,37,38,39,40,41,42,43,44,45,46]. Different applications have been envisaged, such as FSSs [17,18,19,21,23,24,27,32,33,36,38,39,40,42], filters [20,22,26,30,37,43,46], antennas [25,28], and sensors [29,31,34,35,41,44,45]. All these applications are motivated mainly by the huge importance of new telecommunication systems, particularly mobile communication networks, with emphasis on the recently deployed 5G systems and the closed associated Internet-of-Things (IoT). New ideas and perspectives are being explored to further develop these new types of structures. However, having been worked on for about ten years, the topic is already sufficiently mature to justify the publication of a review paper. Therefore, the goal of this paper was to precisely present a review on the work conducted in the field of planar printed structures based on Matryoshka geometries. The paper is organized into seven sections. After this introductory section, Section 2 deals with the printed planar Matryoshka geometry. The Matryoshka cell is described, and the corresponding working principles are analyzed. Section 3, Section 4, Section 5, and Section 6 are dedicated to the description of the use of this type of cell in FSSs, filters, antennas, and sensors in already-reported applications, respectively. At the end, Section 7 contains the main conclusions and the perspectives of present and future developments.

## 2. The Matryoshka Geometry

The Matryoshka geometry is based on concentric rings. As shown in Figure 2, Matryoshka geometries have been conceived as an evolution of the split ring resonators (rings). Starting with a set of rings (so far homothetic), a gap is introduced in each one and then the consecutive rings are connected near the gaps. However, differently from the SRR [47], the rings are connected. As in an SRR, the rings may take different shapes, from simple ones (as square or circular) to other, more complex, canonical or non-canonical geometries. Matryoshka geometries have been implemented in printed circuit board (PCB) technology, both with and without a ground plane. As the SRR, they can be formed by metal strips or by slots in the metal (complimentary configurations). As complimentary configurations, Matryoshka geometries have been used in defected ground structures (DGSs) [30,48] and FSSs [24].

For a specific type of Matryoshka geometry, there are two sub-types: the open and the closed. It is open when there is a gap in the smaller inner ring (Figure 3). As is detailed in the next sections, this gap has a remarkable effect on the structure’s characteristics, namely, on its resonance frequencies. Due to the metal continuity, in the closed configuration, for the first resonance
L_ef_ ≈ λ_refclose_,(1)
whereas, for the open one,
L_ef_ ≈ λ_refopen_/2,(2)
where L_ef_ is the effective length of the structure and λ_ref_ is the effective wavelength [19] for the first resonance. Naturally, there are other (higher-order) resonances. This difference in the behavior of the closed and the open structure can be explained by the continuity required by the closed structure and the interference standing wave pattern imposed by the reflection at the gap of the open structure’s inner ring. This means that, for structures with the same dimensions, the open structure has a first resonance frequency that is approximately half the one of the closed structure. In other words, for the same first resonance frequency, the open structure has an equivalent electrical length that is approximately half of the one of the closed structure, meaning a much more effective miniaturization capability.

The Matryoshka configurations are highly meandered, and the total area occupied is defined by the external ring. The physical parameters of an open square Matryoshka configuration, with four rings, are indicated in Figure 4. For the closed Matryoshka geometry, there is no gap at the inner ring (s = 0). When a Matryoshka geometry is used in an FSS, it is also necessary to specify the unit-cell size.

The average perimeter of the closed geometry (P_N_), corresponding to the physical length defined at the middle of each segment, can be obtained using Equation (3). For the open geometry, s must be subtracted from P_N_.
(3)$$P_{N}=2\left[\sum_{n=1}^{N}\left(L_{xn}+L_{yn}\right)+\sum_{n=1}^{N-1}L_{cn}-2Nw-\left(N-1\right)g\right]$$

In printed planar structures, it is also necessary to specify the substrate characteristics (ε_r_—relative electric permittivity, h—thickness, and tanδ—loss tangent) and the presence or absence of the ground plane. In both cases, the structures are transversally non-homogeneous, and an equivalent homogenous medium can be conceived. For the commonly used substrates, with normal magnetic behavior, an effective relative electric permittivity (ε_ref_) and an effective wavelength (λ_ef_) can be defined.
(4)$$\lambda_{ef}=\frac{\lambda_{0}}{\sqrt{\varepsilon_{ref}}}$$
where λ_0_ is the wavelength in a vacuum. The procedure used to calculate ε_ref_ depends on the type of configuration used, which is associated with the envisaged application. For filters, a microstrip structure has been used, whereas for FSSs, a simple substrate without a ground plane has been selected. For antenna applications, so far, only microstrip structures with DGSs have been employed.

There are some features of the Matryoshka geometries that depend on the specific type of structure and application. These specific features will be detailed in the next sections, where applications as FSSs, filters, antennas, and sensors are analyzed. However, there are some features that are intrinsic of the Matryoshka geometries and therefore are common to all type of applications. These common features re analyzed here using microstrip filters as application examples.

There are different formulas to obtain the ε_ref_ of a microstrip line. A simple non-dispersive model, valid for low frequency, is given in Equations (5)–(7) [49].
(5)$$\varepsilon_{ref}=\frac{\varepsilon_{r}+1}{2}+\frac{\varepsilon_{r}-1}{2}{\left(1+10\frac{h}{w}\right)}^{-ab}$$
(6)$$a=1+\frac{1}{49}ln\left[\frac{{\left(\frac{w}{h}\right)}^{4}+{\left(\frac{w}{52h}\right)}^{2}}{{\left(\frac{w}{h}\right)}^{4}+0.432}\right]+\frac{1}{18.7}ln\left[1+{\left(\frac{w}{18.1h}\right)}^{3}\right]$$
(7)$$b=0.564{\left(\frac{\varepsilon_{r}-0.9}{\varepsilon_{r}+3}\right)}^{0.053}$$

For a 2.0 mm wide microstrip line printed on a FR4 substrate with ε_r_ = 4.4 and h = 1.5 mm, Equation (5) leads to ε_ref_ = 3.23.

The outline of a microstrip filter, with an open Matryoshka geometry of two rings, is shown in Figure 5. The input and output microstrip lines are 2.8 mm wide (50 Ohm characteristic impedance). W = 2.0 mm and g = s = 1.0 mm are used.

The four configurations, indicated in Table 1, were numerically simulated in Ansoft HFSS [50]. Square rings were used (L_n_ = L_xn_ = L_yn_). All the four configurations had the same average perimeter (P_N_ = 178.00 mm).

The simulated transmission coefficient of the four configurations is shown is Figure 6 for the open configuration and Figure 7 for the closed one.

As can be concluded from Figure 6, the open configurations present adequate characteristics for a stopband filter, that is, high attenuation in the stopband, low attenuation in the passband, steep slope transition from passband to stopband (and vice-versa), and large bandwidth. However, that is not the case for the closed configurations (Figure 7). The open configuration provides a higher order filter because it offers two different resonance paths and the closed configurations just one. Moreover, as predicted, the open configurations have much lower first resonance frequencies. As is verified in the next sections, this conclusion was also obtained for the Matryoshka configurations used for other envisage applications (FSSs, antennas, sensors). Table 2 summarizes the characteristics of the open Matryoshka filter configurations.

For a single ring configuration, the effective length can be simply calculated as the average perimeter. However, for multiring Matryoshka configurations, there is coupling between the rings, and there is not a simple physical interpretation of the effective length. To pre-design two rings’ configurations, curve fitting was used to obtain L_ef_ associated with the first two resonances [20].
(8)$$L_{ef1}=2\left[3\left(L_{1}+L_{2}-4w\right)+L_{c}\right]$$
(9)$$L_{ef2}=2\left(3L_{1}+2L_{2}-10w\right)$$

Equation (2) tends to provide a better estimation of the first resonance frequency for intensive coupling (small Lc). Although the relative error can reach about 12% (for the first resonance of configuration 4), Equation (2) is still very useful at the pre-design stage of these filters. These configurations provide miniaturized filters with very large bandwidth. The four configurations have different sizes for the rings and separation between them, but, because they have the same perimeter, the stopband characteristics of the open configurations are very similar.

Microstrip filters based on open Matryoshka geometries with two, three, and four rings were also simulated. The corresponding dimensions are indicated in Table 3. The previously indicated FR4 substrate, w = 2.0 mm and g = s = 1.0 mm, were used, again.

The obtained |S21| results are shown in Figure 8.

Table 4 contains the main simulation results associated with the first two resonances shown in Figure 8.

The use of more rings leads to the appearance of more resonances and, if the average perimeter increases, to a decrease in the frequency associated with the first two resonances.

The surface current density, on configuration 6, at the resonance frequencies and for frequencies between them, is shown in Figure 9.

There was a common pattern of the surface current distribution at the resonance frequencies. Being a stopband filter, there was no transmission at the resonance frequencies. In fact, for such frequencies (Figure 9b,c,e,g), the current at the output port was negligible, and the current at the input port was very strong due to a positive interference of the incident wave and the waves reflected at the two parallel paths, mostly if there was a good input impedance matching. For the frequencies between resonance frequencies (Figure 9a,d,f), there was almost perfect transmission. It was also noticeable that the current magnitude on the inner rings increased as frequency went up.

## 3. Examples of Application as FSS

The first application suggested for the Matryoshka geometry was as FSS [17,18]. This is quite logical since, at the time, there was already a strong and continued research activity in the topic of FSS at GTEMA, and FSS was already widely used in microwave, millimeter wave, and Terahertz frequency bands. There are many specific applications of FSSs, such as RFID, absorbers, rasorbers, reconfigurable intelligent surfaces (RIS), RF energy harvesting, polarizers, dichroic sub-reflector and reflector antennas, reflectarray antennas, beam-switching antennas, lens antennas, and radio astronomy [51,52,53]. A very important initial choice in the design of an FSS is the geometry of the unit-cell. Despite the variety of available geometries, with the rapid growing of wireless technologies, telecommunication system requirements impose a continuing challenge to meet characteristics such as miniaturization, multiband operation, and polarization independence.

### 3.1. Closed Matryoshka FSSs

In [17,18], the closed square Matryoshka geometry with two rings, shown in Figure 10, was introduced. A 0.97 mm thick FR4 substrate with ε_r_ = 4.4 and tanδ = 0.02 was used. The dimensions (in mm) of two configurations are indicated in Table 5, and Wx = Wy = 24.0 mm.

Using Equations (1), (4), and (5) and the procedure proposed in [17,18] to estimate L_ef_ and ε_ref_, the initial dimensions of the Matryoshka unit-cell, fulfilling the specifications, can be obtained. L_ef_ depends on the polarization considered. Usually the two orthogonal linear polarization (vertical-V and horizontal-H) are employed. The use of a numerical simulator can then provide the necessary optimization.

The simulation and experimental transmission coefficient results, obtained for configuration 1, are shown in Figure 11. The simulation results correspond to an infinite FSS (Floquet boundary conditions [51,52]). The horn antennas, available at the time, only allowed measurements above 4.5 GHz.

A general good agreement was obtained between the simulation and experimental results. The ripple in the experimental results was caused by the reflections on the objects present in the non-anechoic environment of the laboratory. The transmission coefficient results depended on the polarization (horizontal-H or vertical-V) of the incident electric field. However, the first resonance frequency was the same for both polarizations (1.75 GHz), but the −10 dB bandwidth was larger for the vertical polarization (19.8% and 10.0%).

The simulation results obtained for the two configurations are compared in Figure 12.

Although both configurations had the same external ring dimensions, configuration 1 had a larger L_ef_, and consequently its resonance frequencies were lower. For instance, f_res11_ = 1.75 GHz and f_res12_ = 2.06 GHz. Configuration 2 had a much larger bandwidth for both polarizations. For the vertical polarization, BW_1_ = 19.8%, whereas BW_2_ = 26.7%. The dimensions of the internal ring can be used to fine tune the FSS and to control the bandwidth.

### 3.2. Open Matryoshka FSSs

In [19,32], the open Matryoshka geometry was introduced, reducing the first resonance frequency to approximately half, when compared to the closed Matryoshka geometry FSS, previously analyzed. It is interesting to compare, now for application in FSSs, the simple rings, with closed and open Matryoshka geometries, as shown in Figure 13. Again, a 0.97 mm thick FR4 substrate with ε_r_ = 4.4 and tanδ = 0.02 was used. A unit-cell with size Wx = Wy = 24.0 mm, L_1_ = 22.0 mm, L_2_ = 12 mm, L_c_ = 3.5 mm, w = 1.5 mm, and g = s = 1.0 mm was chosen.

The simulated |S21| results are shown in Figure 14 for horizontal polarization, and in Figure 15 for vertical polarization.

The simple rings configuration was physically symmetric and therefore its response was the same for the horizontal and vertical polarizations. This was also the case for the closed Matryoshka configuration, but only for the first resonance. However, the open Matryoshka configuration had completely different responses for the two polarizations. Although the three configurations occupied the same area, the two unit-cell with Matryoshka geometry provided much lower first resonance frequencies, especially for vertical polarization. The results associated with the first resonance are summarized in Table 6.

The FSS with open Matryoshka geometry provided a remarkable reduction of the first resonance frequency, especially for vertical polarization, when compared with the closed configuration (43%) and with the simple rings (61%). However, there was a substantial reduction of the bandwidth.

The simulation results for the open Matryoshka configuration were validated with experimental results, as shown in Figure 16.

More rings can be used to increase the effective length and therefore further reduce the first resonance frequency, increase the number of resonances, and provide a fine-tuned control of the resonances and of the bandwidth [19,32]. To confirm these conclusions, an FSS with an open Matryoshka unit-cell with three rings was designed, fabricated, and tested. Again, a 0.97 mm thick FR4 substrate with ε_r_ = 4.4 and tanδ = 0.02 was used. A unit-cell with size Wx = Wy = 24.0 mm, L_1_ = 22.0 mm, L_c1_ = L_c2_ = 2.25 mm, L_2_ = 14.5 mm, L_3_ = 7.0 mm, w = 1.5 mm, and g = s = 1.0 mm was chosen. Photos of the prototype and of the experimental setup are shown in Figure 17.

The simulation and experimental results of the three rings of open Matryoshka geometry for vertical and horizontal polarizations are shown in Figure 18. The experimental results were only able to be measured starting at 1 GHz. There was a general good agreement between the simulation and experimental results. As mentioned before, and as can be verified in Figure 17b, the ripple in the experimental results was caused by the reflections on the objects present in the non-anechoic environment of the laboratory.

The simulation results of the FSS with open Matryoshka unit-cells with two and three rings are summarized in Table 7.

It was confirmed that, for the same dimension of the external ring, the increase in the number of rings (two to three) provided a reduction of the first resonance frequency, an increase in the number of resonances, and fine tune control of the resonances. However, a reduction of the bandwidth was verified.

### 3.3. Polarization of Independent FSSs

To overcome the inconvenient polarization dependence, analyzed in the previous section, a new configuration of the Matryoshka geometry was proposed in [21,36]. As shown in Figure 19, this new type of configuration has been conceived as an evolution from the simple circular ring keeping the main Matryoshka characteristics, that is, the area occupied is defined by the external ring only, and more rings can be added internally, maintaining electrical continuity. The physical characterization of the circular Matryoshka geometry is defined in Figure 20.

The FSS unit-cells with the three geometries shown in Figure 19 were designed, fabricated, and tested [21,36]. A 0.762 mm thick FR4 substrate with ε_r_ = 4.4 and tanδ = 0.02 was used. An FSS with 10 × 10 unit-cells with size Wx = Wy = 20.0 mm and w = g = 0.8 mm was chosen. The radius of the unit-cells’ rings are indicated in Table 8. The radius reduction rate was maintained from ring to ring.

As the FSSs were horizontally and vertically symmetric, they were polarization independent. Therefore, for these three cases, only results obtained for vertical polarization are shown. An important characteristic of an FSS is its sensitivity to the angle of the incident electromagnetic wave. Four angles of incidence were considered, from normal incidence (θ = 0) to θ = 45°, with a 15° interval.

In [21,36], the formulas indicated in Equations (10) to (12) are proposed to estimate the first resonance frequency of the FSS with one, three, and five rings, respectively. These formulas were used to specify the radius of the three configurations at the initial stage of the design procedure.
(10)$$f_{rFSS1}=\frac{3\times 10^{8}}{2\pi r_{1}\sqrt{\varepsilon_{ref}}}$$
(11)$$f_{rFSS2}=\frac{3\times 10^{8}}{2\pi \left(r_{1}+r_{3}\right)\sqrt[{}]{\varepsilon_{ref}}}$$
(12)$$f_{rFSS3}=\frac{3\times 10^{8}}{2\pi \left(r_{1}+r_{3}+r_{5}\right)\sqrt[{}]{\varepsilon_{ref}}}$$
where ε_ref_ is the effective relative permittivity of the equivalent homogeneous structure [21,36].

The prototypes, shown in Figure 21, were fabricated, and they were measured using the setup shown in Figure 22.

The simulation and experimental results obtained for the transmission coefficient of the three prototypes are shown in Figure 23, Figure 24 and Figure 25. The simulation results for the four different incident angles were very similar and, therefore, for the sake of clarity, only one curve is represented.

There was a good agreement between the numerical simulations and experimental results for the three prototypes. In general, the measured results were below the simulation ones. This difference was about 5 dB, on average, and tended to increase as the oblique angle of incidence increased. This effect may have been caused by the finite size of the window used in the measurement setup (Figure 22).

As the perimeter of the FSS unit-cell increased with the number of rings, more resonances appeared. The results, for the first resonance frequency, are compared in Table 9.

There was a good agreement between the numerical simulation and experimental results. Moreover, also the estimation provided by Equations (10)–(12) was accurate enough for the initial stage of the design (relative error below 5%). It is clear that the three prototypes had low sensitivity from the inclination angle and that there was a remarkable reduction in the first resonance frequency as the number of rings increased (44% from FSS1 to FSS3).

Recently a polarization-insensitive miniaturized multiband FSS with Matryoshka geometry elements was proposed [42]. From an initial polarization-sensitive unit-cell with a single element, there was an evolution to a combination of four orthogonals of such unit-cells (Figure 26).

The simulation results for the transmission coefficient of the two Matryoshka unit-cells, for normal incidence (θ = 0), are shown in Figure 27 and Figure 28.

The single-element unit-cell had a strong polarization dependence, but the four orthogonal elements unit-cell was almost perfectly polarization independent. Moreover, from the results presented in Figure 29 and Figure 30, it can be concluded that the new four elements arrangement also provided low sensitivity to the angle of incidence. For horizontal polarization, the curves for θ = 0, θ = 20°, and θ = 40° were almost coincident; only the curve for θ = 60° was slightly different. For vertical polarization, the situation was almost the same, but both the curves for θ = 40° and for θ = 60° were slightly different from the other two.

### 3.4. Combination of an FSS with Dipoles

One of the advantages of the Matryoshka geometry is that it can be combined with other geometries in order to obtain an FSS with low coupling between the fields of each geometry, allowing for the control of the respective frequency responses. This is particularly interesting for the design of multiband FSS. In [23,27], cross-dipoles and Matryoshka geometries were combined to achieve a polarization-independent, triple-band FSS. The combined geometries are shown in Figure 31. The prototype, shown in Figure 32, was fabricated and characterized.

In [23], a 1.6 mm thick FR4 substrate with ε_r_ = 4.4 and tanδ = 0.02 was used. The simulation results shown in Figure 33 correspond to an FSS with 5 × 5 unit-cells with size Wx = Wy = 40.0 mm, w = 1.5 mm, and g = 1.0 mm. Moreover, L_x1_ = L_y1_ = 24.0 mm, L_x2_ = L_y2_ = 19.0 mm, L_x3_ = L_y3_ = 14.0 mm, d_x1_ = d_y1_ = 15.0 mm, d_x2_ = _dy2_ = 8.5 mm, d_x3_ = d_y3_ = 6.0 mm, and L_dip_ = 39.0 mm. Three resonances can be observed (f_r1_ = 1.81 GHz, f_r2_ = 2.43 GHz, and f_r3_ = 3.19 GHz). They corresponded to the superposition of the first and second resonances of the Matryoshka geometry (f_r1_ = 1.82 GHz and f_r2_ = 3.18 GHz) with the first resonance of the cross-dipoles (f_r1_ = 2.46 GHz). The three resonances can be controlled separately, which is a very interesting feature that adds flexibility in the design for different potential applications. For instance, it is possible to design the dipole so that its first resonance frequency is close to one of the resonance frequencies of the Matryoshka geometry (first or second). By doing this, an increase in the bandwidth of the combined resonances can be obtained. Numerical simulation results are compared with experimental results, obtained for different angles of incidence, in Figure 34.

The simulation results for the four different incident angles were very similar and, therefore, for the sake of clarity, only one curve is represented. There was a general good agreement between the simulation and experimental results, which validates the design procedure. Moreover, the angular stability was confirmed. The discrepancies observed may have been caused by the finite size of the window used in the measurement setup (Figure 32b).

### 3.5. Complimentary FSSs

The Matryoshka geometry, described in the previous section, is also used in its complementary form [24,39]. The FSS unit-cell is obtained as described in Figure 35.

It was shown, in the previous section, that the Matryoshka geometry with metal strips had a stopband response associated with its resonances. The complimentary Matryoshka geometry had a passband behavior. Two prototypes of these complimentary Matryoshka configurations with 9 × 9 unit-cells, each cell with 22.4 × 22.4 mm^2^, were designed, fabricated, and tested [24,39]. A 1.6 mm thick FR4 substrate with ε_r_ = 4.4 and tanδ = 0.02 and w = g = 1.0 mm was used. The simulation results shown in Figure 36 correspond to FSS1 (L_x1_ = L_y1_ = 20.4 mm, L_x2_ = L_y2_ = 16.4 mm, L_x3_ = L_y3_ = 12.4 mm, d_x1_ = d_y1_ = 11.4 mm, d_x2_ = d_y2_ = 7.4 mm, d_x3_ = d_y3_ = 5.5 mm) and FSS2 (L_x1_ = L_y1_ = 15.4 mm, L_x2_ = L_y2_ = 11.4 mm, L_x3_ = L_y3_ = 7.4 mm, d_x1_ = d_y1_ = 9.0 mm, d_x2_ = d_y2_ = 5.0 mm, d_x3_ = d_y3_ = 3.0 mm). These dimensions were calculated using the design formulas proposed in [24] to provide passbands centered at 1.5 GHz and 3.5 GHz for FSS1 and 2.5 GHz and 5.1 GHz for FSS2, as well as a stopband centered at 2.45 GHz for FSS1 and 3.5 GHz for FSS2. Photos of these complimentary FSS prototype are shown in Figure 37.

The results shown in Figure 36 demonstrate that it is possible to adjust the range of the stopbands and passbands according to the specifications.

As shown in the previous section, this Matryoshka configuration has very good angular stability. From the results presented in Figure 38 and Figure 39, it can also be concluded that the complimentary Matryoshka configuration presents the same good angular stability.

The simulation results for the four different incident angles were very similar and, therefore, for the sake of clarity, only one curve is represented. Taking into account that the experimental results were obtained in a simple non-anechoic room, there was a good agreement between the numerical simulation and experimental results. The resonance and antiresonance frequencies were confirmed experimentally with relative error differences below 4.5% [24].

### 3.6. Reconfigurable FSSs

For some applications, reconfigurability is a very attractive feature for an FSS. In this case, combined geometries can be useful. Adding a PIN diode between the vertical dipoles of the FSS presented in [23], and analyzed in Section 3.4, a reconfigurable FSS (RFSS) was obtained, as described in [27]. Additionally, as shown in Figure 40, a RF inductor was added between the horizontal dipoles to act as an RF choke [49]. The RFSS prototype, shown in Figure 40c, was fabricated and characterized [27].

A 1.6 mm thick FR4 substrate with ε_r_ = 4.4 and tanδ = 0.02 was used. The prototype shown in Figure 40c corresponds to an FSS with 7 × 7 unit-cells with size W_x_ = W_y_ = 30.0 mm, w = 1.5 mm, and g = 1.0 mm. Moreover, L_x1_ = L_y1_ = 24.0 mm, L_x2_ = L_y2_ = 19.0 mm, L_x3_ = L_y3_ = 14.0 mm, d_x1_ = d_y1_ = 15.0 mm, d_x2_ = d_y2_ = 8.5 mm, d_x3_ = d_y3_ = 6.0 mm, and L_dip_ = 29.0 mm. The numerical simulation and measured results are presented in Figure 41, Figure 42, Figure 43, Figure 44 and Figure 45. Figure 41 corresponds to unit-cells without PIN diodes and without inductors. These results serve as a reference. Figure 42 corresponds to unit-cells with PIN diodes but without inductors. Figure 43, Figure 44 and Figure 45 correspond to unit-cells with both PIN diodes and inductors.

In Figure 41, the simulation results for the horizontal and vertical polarization are identical; for the sake of clarity, only one curve is represented. There was a reasonable agreement between simulation and experimental results. The experimental second and third resonances were substantially deviated from the simulations. The difference was able to reach 6.6% and was mainly caused by the non-anechoic environment of the laboratory and eventual fabrication inaccuracies.

As shown in Figure 42, the agreement between numerical simulation and experimental results was good, except for the second resonance and horizontal polarization (7% relative error). In addition to the already mentioned general error causes (non-anechoic environment and fabrication inaccuracies), there must be some other problem not detected. However, as this configuration is just an intermediate step, and new prototypes would be fabricated for the next steps, the work was continued.

The results shown in Figure 43 indicate that, as expected, the state of the diode did not affect the horizontal polarization. Moreover, the problem associated with the second resonance frequency, detected in Figure 42, disappeared. The relative error for the second resonance frequency was then only about 4%. It was, therefore, verified that there is the need to use the inductors. As shown in Figure 44, a good agreement between the simulation and experimental results was verified for both OFF and ON states.

As can be concluded from the results presented in Figure 45, for vertical polarization, the reconfiguration of the FSS was effective, with a reconfigurable bandwidth of 0.37 GHz (17%), from 2.03 GHz to 2.40 GHz, with at least 10 dB of difference between OFF and ON bias states of the diodes.

## 4. Examples of Application as Filter

Providing access to telecommunication networks in the most diverse locations, with quality of service and without losing mobility, poses major challenges for manufacturers of mobile equipment and infrastructures. In both cases, filters play a fundamental role, separating the desired signals from the unwanted ones. Telecommunication systems, namely, 5G wireless communications, require filters with operating conditions that are increasingly challenging in terms of frequency response, in addition to low cost, weight, and volume (miniaturization). In this sense, new microwave filter configurations have been developed [54,55]. To meet these requirements, planar filters are widely used. Planar filters can be viewed as resonators, lumped or quasi-lumped, for which the resonance frequency is determined by the geometry [49,56]. There are other important applications of filters, such as Wi-Fi, global satellite navigation systems, test, and measurement equipment (spectrum and network analyzers) [53]. Aiming to take advantage of the characteristics observed for the Matryoshka geometry when used in FSSs (miniaturization and multiband operation), filters based on Matryoshka geometries were introduced in [20,26].

### 4.1. Filters with a Square Matryoshka Geometry

Printed planar microstrip filters based on open Matryoshka square geometries were presented in [20,37]. Filters with two and three rings are shown (Figure 46).

The physical characteristics of the five configurations chosen are described in Table 10. The definition of these physical characteristics is provided in Figure 4. A 1.5 mm thick FR4 substrate with ε_r_ = 4.4 and tanδ = 0.02 was used. The input and output microstrip lines were 2.8 mm wide (50 Ohm characteristic impedance); W = 2.0 mm and g = s = 1.0 mm were used. For the three ring configurations, L_c_ = L_c1_ = L_c2_.

Photos of the fabricated prototypes are shown in Figure 47.

As explained in Section 2, a design procedure was developed to estimate the initial dimensions of the filter that fulfill the specifications. Comparisons of the numerical simulation and experimental results are shown in Figure 48 (for config1, config2, and config3) and Figure 49 (for config4 and config5).

Figure 48 shows a very good agreement between the simulation and experimental results. Table 11 summarizes the experimental characteristics of the three initial configurations.

As shown in Figure 49, a very good agreement between the simulation and experimental results was also obtained. Table 11 also summarizes the experimental characteristics of the two remaining configurations.

It can be concluded that the first two resonance frequencies (f_r1_ and f_r2_) depended on the perimeter of the rings. For the same external dimensions (L_1_), the inner rings can be used to have a fine control of the stopband frequency range and of the bandwidth.

### 4.2. Filters with a Circular Matryoshka Geometry

Filters based on open Matryoshka circular ring configurations were studied in [26,57]. The physical characteristics of a one-ring configuration are defined in Figure 50.

Five configurations of this type of stopband filter were studied in [26] (Table 12). A 1.52 mm thick Rogers RO3003 substrate with ε_r_ = 3.0 and tanδ = 0.001 was used. The input and output microstrip lines (P1 and P2) were 3.8 mm wide (50 Ohm characteristic impedance). W = 1.0 mm and g = s = 1.0 mm were used.

Photos of the fabricated prototypes are shown in Figure 51.

Similarly to the square Matryoshka configurations, a design procedure was developed to estimate the initial dimensions of the filter that fulfill the specifications. A comparison of the numerical simulation and experimental results is shown in Figure 52 (for config1, config2, config3, and config4) and Figure 53 (for config1 and config5).

There was a very good agreement between the numerical simulation and the experimental results, as shown in Figure 52. Table 13 summarizes the experimental characteristics of the five configurations. The first resonance frequency was almost independent of the second ring, but the second resonance frequency and the bandwidth increased substantially as the radius of the second ring decreased.

As shown in Figure 53, a very good agreement between the simulation and experimental results was also obtained. A general conclusion, in line with the analysis carried out on the filters (and also the FSSs) with square Matryoshka geometry, is that, keeping the external dimension of the structure, the resonance frequency decreased (higher miniaturization) when more rings were used (higher meandering), but the bandwidth decreased.

### 4.3. Filters with a DGS

A DGS was formed by removing a small part from the metallic ground plane in planar printed circuit boards, most frequently in microstrip lines, as shown in Figure 54.

Due to ease of integration, design flexibility, and compactness, DGSs have found several applications such as in planar antennas [58,59], filters [60,61], power dividers [62,63], sensors [64,65], and wireless power transfer [66,67]. A DGS based on a Matryoshka geometry, as shown in Figure 54, was introduced in [30,43,68]. In [30], a method to design this type of DGS based on simple formulas is proposed. Four configurations were designed, fabricated, and tested [30]. A 1.6 mm thick FR4 substrate (ε_r_ = 4.4 and tanδ = 0.02) was used. The corresponding dimensions are indicated in Table 14. The definition of the dimensions is provided in Figure 4.

Photos of the front and back sides of the prototypes are shown in Figure 55. Figure 56 provides the comparison of simulation and experimental |S21| results of these prototypes.

Good agreement is shown in Figure 56 between the numerical simulation and experimental results. The tendency of the resonance frequency was the same as the metal Matryoshka configuration, that is, as the area of the structure decreased, the resonance frequency increased.

To assess the capabilities of the Matryoshka geometry to perform as a DGS, a comparison with a DGS of the common dumbbell geometry is presented in [30]. To have a fair comparison, the square dumbbell geometry had the same area as the Matryoshka geometry. The simulation results for the four configurations indicated in Table 14 are presented in Figure 57.

A summary of the results shown in Figure 57 is provided in Table 15. The resonance frequency of the Matryoshka geometry was much lower than the resonance frequency of the dumbbell geometry (larger miniaturization), but the bandwidth was much narrower.

### 4.4. Filters with a DGS and a Dielectric Resonator

Very recently, a compact filter combining a Matryoshka geometry DGS with a high-permittivity dielectric resonator was proposed [43]. The purpose was to improve the frequency response characteristics, mainly selectivity, and miniaturization. A prototype was designed, fabricated, and tested. A 1.6 mm thick FR4 substrate (ε_r_ = 4.4 and tanδ = 0.02) was used. As shown in Figure 58, a calcium cobaltite disk (ε_r_ = 90) with a diameter of 10.0 mm and a thickness of 1.9 mm was inserted into config3 of the previous section, below the ground plane, in contact with the DGS. A photograph, with a bottom view of the prototype, is shown in Figure 59. The filter transmission coefficient was simulated and measured for different positions of the dielectric disk. The corresponding results, for the disk centered on the DGS Matryoshka square geometry center, and on the DGS Matryoshka square geometry corner, are shown in Figure 60. The results obtained for the filter without a dielectric disk are also shown, for reference.

A very good agreement between the numerical simulation and experimental results was obtained. The use of a dielectric resonator can provide further miniaturization of the structure. The experimental resonance frequency moved from 2.939 GHz (no disk) to 1.849 GHz (center) and to 1.322 GHz (corner), which corresponded to 36.8% and 55.0% reductions, respectively. Again, the price to pay was the reduction of bandwidth, which was 58.5% (center) and 86.2% (corner). The position of the dielectric disk can be used to fine-tune the central frequency of the filter’s response.

## 5. Examples of Application as Antenna

Due to their inherent multiresonant characteristics, Matryoshka geometries are suitable for multiband and/or wideband antenna configurations [25,28]. Moreover, because of the meandering of the nested rings, they have also been used to provide antenna miniaturization [25,28]. These features can be advantageously combined with printed antennas in general and microstrip patch antennas in particular [25,28] to be used in small mobile communication terminals and mass production electronic gadgets. Microstrip is one of the most successful antenna technologies. Such success stems from well-known advantageous and unique properties, such as a low profile, light weight, planar structure (but also conformal to non-planar geometries), mechanical strength, easy and low-cost fabrication, easy integration of passive and active components, easy combination to form arrays, and outstanding versatility in terms of electromagnetic characteristics (resonance frequency, input impedance, radiation pattern, gain, polarization). Microstrip patch antennas can be used in a very wide frequency range, extending roughly from about 1 GHz to 100 GHz [69]. So far, the Matryoshka geometries have been used in microstrip patch antennas to modify the ground plane and implement it as a DGS [25,28]. DGSs have been used in microstrip antenna implementations to provide multiband and/or wideband behavior, improve gain and cross-polarization, and suppress higher order modes and mutual coupling (in arrays) [70,71,72,73,74]. Many different shapes of the DGS slots have been used, ranging from canonical geometries (rectangular, triangular, circular) to non-canonical (H-shaped, dog bone-shaped) [75]. Recently, such variety was enhanced with the Matryoshka geometry [25,28].

In [25], a comparison of the performance of a microstrip patch antenna with a DGS ground plane with circular SRRs [76] and Matryoshka geometries is presented. The emphasis was on the open Matryoshka configuration. In [28], a detailed comparative analysis of the performance of open and closed Matryoshka DGS geometries was carried out. In all the cases, a cheap FR4 substrate with relative electric permittivity of 4.4, thickness of 1.6 mm, and loss tangent of 0.02 was used.

### 5.1. Reference Microstrip Patches

In [25], as an application example, the dimensions of a rectangular patch were chosen to provide the first resonance at 2.5 GHz. The initial dimensions obtained with the transmission line method [77] were optimized using the ANSYS Electronics Desktop 2018.1.0, release 19.1.0 [78]. A patch width (W) of 37.0 mm, length of (L) 27.8 mm, and a square ground plane with a 53.0 mm side were chosen. The patch was fed with a 2.8 mm wide microstrip transmission line (50 Ohm characteristic impedance) and inset that is 1 mm wide and 8 mm long. The corresponding simulated input reflection coefficient is shown in Figure 61, for reference.

The first resonance (2.52 GHz), the second resonance (3.86 GHz), and the third resonance (4.74 GHz) were well matched to the 50 Ohm feed microstrip transmission line. To validate the design procedure used, a protype of the microstrip patch antenna was fabricated using a conventional photolithography technique. The amplitude of the experimental input reflection coefficient (|S_11_|), also shown in Figure 61, was measured with an Agilent E5071C (Agilent, Santa Clara, CA, USA) vector network analyzer. Taking into account that the FR4 substrate used is low cost, and its characteristics are only generically known, there was a good agreement between the numerical simulations and experimental results. For the frequency of interest (first resonance), the difference in the frequency was only 3.4% (88 MHz), and the |S_11_| level was almost the same (−33 dB). This antenna presented the usual almost hemispherical broadside radiation pattern [77] with a gain of 6.18 dBi at 2.52 GHz.

Another microstrip patch was designed so that using a DGS ground plane, the same first resonance (2.5 GHz) of the simple patch, described above, could be obtained [25]. In this case, the patch was designed to have alone the first resonance frequency at 3.5 GHz. The patch and ground plane sizes were 28.0 mm (W), 20.0 mm (L), and 38.0 mm, 45.0 mm, respectively. The corresponding simulation and experimental input reflection coefficient are shown in Figure 62.

The difference in the simulation and experimental resonance frequencies was only 2.0% (69 MHz), and the |S_11_| level was below −26 dB, for both curves.

### 5.2. DGS Uni-Cell

From initial exploratory simulations [25], it is concluded that the patch with a DGS would present the first resonance frequency at 2.5 GHz when the DGS unit-cell alone had the first resonance at about 2.6 GHz. Therefore, both the complementary open Matryoshka and circular SRR configurations were designed to provide such a 2.6 GHz first-resonance frequency. To take into account the intrinsic characteristics of the unit-cells, their analysis was performed by considering an infinite FSS with 20 × 20 mm^2^ unit-cells. The two configurations are shown in Figure 63. Ansys HFSS [78] was used for the simulations.

The complementary open square Matryoshka configuration had dimensions of L_1_ = 6.8 mm and L_2_ = 4.8 mm. For the circular complementary SRR, r_1_ = 5.4 mm and r_2_ = 3.5 mm was used. In both cases, the trace and slot widths were 0.50 mm and 0.25 mm, respectively. The simulated |S21| results are shown in Figure 64.

It can be concluded that, as required, both unit-cells provided the first resonance frequency at about 2.6 GHz.

### 5.3. Patch with DGS

The patch’s ground plane was changed by the introduction of a DGS with an open square Matryoshka and a circular SRR [25], as shown in Figure 65.

The simulation and experimental results for the amplitude of the input reflection coefficient, for both DGS unit-cell geometries, are shown in Figure 66.

There were some differences between the simulation and experimental results. For the first resonance frequency, the experimental result for the SRR geometry was 8.8% (218 MHz) below the simulation one, whereas for the Matryoshka geometry, the experimental result was 6.7% (165 MHz) above the simulation. These differences were mainly caused by the inaccuracy of the fabrication process, mostly related with the narrow (0.25 mm) slots. However, these unwanted differences did not jeopardize the envisage proof of concept, that is, both the DGS configurations provided a remarkable miniaturization of about 46% (in the area of the microstrip patch).

The farfield radiation patterns of the patch with the two DGS configurations are shown in Figure 67, Figure 68 and Figure 69.

When compared with the radiation pattern of the common patch, the main difference was the high radiation level below the ground plane. In contrast with the typical hemispherical type of radiation pattern observed for the common microstrip patch [77], a bi-hemispherical type of radiation pattern was caused by the DGS. This was expected, first due to the introduction of slots in the ground plane and second because the slots were near resonance and therefore with enhanced radiation. This type of radiation pattern may be interesting for application where a more uniform spatial radiation power distribution is required. For the DGS with Matryoshka geometry, the maximum gain (4.9 dBi) was obtained in the back hemisphere (θ ≈ 180°). For the DGS with SRR geometry, the direction of maximum radiation was kept on the front hemisphere (θ ≈ 0), with 4.6 dBi gain, but the front-to-back (FBR) ratio was low (2.8 dB). The drop of about 1.4 dB in the gain was related to the more uniform distribution of radiated power in space.

The main radiation characteristics, obtained by simulation, are summarized in Table 16.

A fair comparison of the miniaturization capability of the two DGS geometries under analysis (open Matryoshka and circular SRR) must consider unit-cells with the same dimension. The first resonance frequency of the microstrip patch with a DGS ground plane, as a function of the maximal dimension (the side of the square open Matryoshka geometry and diameter of the circular SRR), is shown in Figure 70.

As can be observed, the open Matryoshka geometry provided much lower simulation results for the frequency of the first resonance. Moreover, the experimental results obtained for the two fabricated prototypes had a reasonable agreement with the simulations (a difference less than 9%). The average difference, for the resonance frequencies associated with each DGS geometry, was almost 1 GHz (0.94 GHz), which corresponded to 35.4%. This proves that the proposed open Matryoshka geometry had a much stronger miniaturization capability than the conventional circular SRR. As verified in Section 3 and Section 4, the same conclusion was obtained for FSS [32] and filter [20] applications.

In a very recent work [28], a detailed analysis of the effects of a DGS with a Matryoshka unit-cell in the ground plane of a microstrip patch was carried out. A complete sensitivity analysis of the influence of the geometric parameters of the unit-cells in the antenna performance was conducted. The main conclusions obtained in [25] were confirmed and were supported by an extensive and systematic analysis, with simulations and experimental validation. The emphasis was on the comparison of the miniaturization capabilities of the open and closed complementary Matryoshka geometries. As an example, some results obtained for an optimized configuration are reproduced below.

The two configurations of the microstrip patch with a DGS ground plane, that is, with open and closed Matryoshka cells, are shown in Figure 71.

The |S21| of a 50 Ohm microstrip line with a DGS with open and closed Matryoshka square geometries (Figure 72) (L_1_ = 7.5 mm, L_2_ = 4.5 mm, w = 0.5 mm, and s = g = 0.5 mm), as a function of frequency, is shown in Figure 73. The measurement setups used are shown in Figure 74.

There was a good agreement between simulation and experimental |S21| results. Although the open and closed Matryoshka unit-cells had the same dimensions, the first resonance of the open structure (2.40 GHz) was 47.4% below the first resonance of the closed structure (4.57 GHz). However, the closed structure had a much wider −10 dB bandwidth (560 MHz compared with 150 MHz). As shown in Figure 75, this reduction led also to a reduction of the first resonance frequency of the patch with the open Matryoshka DGS.

The input reflection coefficient of the common patch (without DGS) is also shown for reference in Figure 75. In this case, the microstrip patch antenna with open Matryoshka DGS presented the first resonance frequency at 2.35 GHz, which was 28.5% below the first resonance of the microstrip patch antenna with closed Matryoshka DGS (3.29 GHz) and 33.2% below the first resonance of the microstrip patch antenna with a solid ground plane (3.52 GHz).

The surface current distribution, shown in Figure 76, provided physical insight into the antenna’s radiation mechanisms. The current distribution on the common patch (without DGS) is also shown, for reference.

As expected, the presence of the DGS enormously changed the current distribution, not only on the ground plane but also on the patch. This change was more effective for the open Matryoshka geometry. The almost constant current distribution along the common patch width was strongly perturbed by the DGS.

The 3D radiation patterns of the patch with a DGS, with closed and open Matryoshka geometries, are shown in Figure 77. Again, the radiation pattern of the common patch (without DGS) is also shown, for reference.

The maximum gains for the common patch, the patch with closed Matryoshka DGS, and the patch with open Matryoshka DGS were 4.24 dBi, 3.55 dBi, and 2.40 dBi, respectively. Naturally, the increase in the back radiation caused by the DGS implied a decrease in the maximum gain, especially for the open Matryoshka DGS.

A summary of the main results obtained for the patch antenna without DGS and with DGS with open and closed Matryoshka geometries is presented in Table 17.

It can be concluded that both Matryoshka DGS geometries provided miniaturization, but the open structure was much more effective. However, both Matryoshka DGS geometries caused a decrease in the gain, being more pronounced for the open structure. The open structure also provided a narrower impedance bandwidth.

## 6. Examples of Application as a Sensor

If a material under test (MUT) is incorporated in a filter and the filter changes its frequency response according to the characteristics of the MUT, this filter can be used as a sensor [79]. Based on this idea, three practical sensors were proposed.

### 6.1. Alcohol Concentration Sensor

A new and simple sensor, based on a microstrip filter with an open Matryoshka configuration, was proposed in [34]. The proposed sensor was designed, fabricated, and successfully applied to detect the alcohol content of a liquid. Photos of the prototype and of the measurement setup are shown in Figure 78. A small acrylic container with internal dimensions 43.7 × 43.7 × 30.0 mm^3^ (57.29 mL capacity) and 3.0 mm and 1.0 mm thick side and bottom walls, respectively, was placed over the filter, centered on the Matryoshka geometry.

The experimental results obtained for the first resonance frequency, as a function of the alcohol concentration, for three different volumes of liquid, are shown in Figure 79.

For a 16 mL volume of the MUT (about 8.4 mm of liquid in the container), the response was almost linear. It was clear that the larger the MUT volume, the better the sensitivity (slope of the curve). Based on the design procedure described in Section 4.1, other prototypes can be fabricated to develop sensors to be used on other frequency bands. Moreover, other liquids can be characterized, based on calibration curves previously validated.

### 6.2. Sucrose Level and Water Content Sensors

In [80], a sensor based on a microstrip filter with a closed Matryoshka geometry DGS was used to obtain the sucrose level of an aqueous solution. Based on the analysis and design procedure described in Section 4.3, the prototype, shown in Figure 80, was developed. It is based on configuration 1 described in Table 14.

A small acrylic container with internal dimensions of 30 × 30 × 15 mm^3^ (13.5 mL capacity) was placed over the filter, centered on the Matryoshka geometry. The filter is used upside down, that is, with the DGS on the top side.

The calibration curve obtained for the determination of the sucrose level in an aqueous solution is shown in Figure 81.

A similar sensor is proposed in [41] to determine the distilled water content in a solution of isopropyl alcohol and distilled water. It is based on configuration 2 described in Table 14. The corresponding calibration curve is reproduced in Figure 82.

### 6.3. Soil Moisture Sensor

In [31,44], a new soil moisture sensor based on a filter with a Matryoshka geometry DGS is described. The filter configuration is represented in Figure 83.

A closed Matryoshka geometry DGS with L_1_ = 20.0 mm, L_2_ = 14.0 mm, w = 2.0 mm, and g = 1.0 mm was used. Photos of the prototype and of the measurement setup are shown in Figure 84.

Two types of soil were measured: a sandy soil usually used in civil construction, with about 98% sand, and a garden soil rich in organic substances. The corresponding experimental results are shown in Figure 85. So far, the resonance frequency has been used, but in this case, the frequency points where |S21| reaches the −6 dB level was used because it is more stable [31,44].

As the water content increased, the sandy soil absorbed less water and the sample saturated more quickly, starting from approximately 24%. On the other hand, garden soil absorbs more water, and the saturation point occurred at about 40%.

## 7. Conclusions and Perspectives of Future Developments

This paper reviewed the research activities developed on the topic of printed planar resonator structures nested inside each other, called Matryoshka geometries. These research activities started about 10 years ago, and most of the work has been performed at the Group of Telecommunications and Applied Electromagnetism (GTEMA) from the Instituto Federal da Paraíba in João Pessoa, Brazil. Although it is still a recent topic, much varied work has been done to describe the different types of structures, discover and explain their properties, obtain physical insight on the working principles, and explore potential applications.

The main characteristics of Matryoshka geometries stem from their meandered nature and the fact that the area occupied is defined by the external ring where the other interconnected rings are nested inside. These characteristics render them strong miniaturization capability and selective multi-resonances that are highly attractive to be used in many microwave devices. The initial application as FSS was quickly extended to other applications, such as filters, antennas, and different types of sensors.

The Matryoshka geometry was described initially, and the closed and open configuration were introduced. The physical parameters that describe the geometry were defined. The main common characteristics were analyzed, and the working principles were studied to provide physical insight on their behavior.

The initial application of Matryoshka geometries, as FSS unit-cells, was fully examined. Closed and open configurations were analyzed in detail, and comparison with simple rings was provided. The sensitivity of the transmission coefficient to each of the physical parameters was evaluated. To overcome the polarization dependence of simple FSS configurations, polarization-independent Matryoshka geometry configurations were introduced and analyzed. The combination of an FSS with a crossed dipole was proposed to enhance the multi-frequency response. Complimentary configurations, where the Matryoshka geometries were implemented with slots, were also analyzed. FSS unit-cells that use PIN diodes to provide reconfigurability were proposed and studied. It was effectively verified that in all the configurations studied, the Matryoshka-geometry-based configurations exhibited the expected multi-resonance behavior, and a remarkable miniaturization was provided.

Microstrip filters that use both square and circular rings with the open Matryoshka geometry were thoroughly examined. Configurations with different dimensions and number of rings were studied. The use of open Matryoshka geometries to implement DGS (in the ground plane) in microstrip filters was also proposed and analyzed. A combination of an open Matryoshka DGS configuration with a dielectric disk resonator was proposed to further enhance the miniaturization capability.

The use of DGS configurations with Matryoshka geometry in the ground plane of microstrip patch antennas was proposed and examined in detail. It was concluded that there was a remarkable miniaturization capability, but a decrease in gain and impedance bandwidth was inevitably observed.

Some applications of the devices based on configurations with Matryoshka geometries have already been envisaged and reported. Microstrip filters with open Matryoshka geometry were used to determine the alcohol concentration present in a liquid solution. The quality of the estimation increased as the volume of the sample increased. A microstrip filter with closed Matryoshka geometry DGS was used in a sensor conceived to obtain the sucrose content level and the distilled water content level of aqueous solutions. The corresponding calibration curves were proposed. Another microstrip filter with close Matryoshka geometry was used to obtain the percentage of moisture in soil samples. These applications of Matryoshka-based geometry configurations in sensors are very promising because they are simple, low cost, and potentially accurate.

Although many configurations based on Matryoshka geometries have been proposed and some potential applications of them as FSSs, filters, antennas, and sensors have been explored, it is clear that much more remains to be investigated. This is natural, since it is just a ten-year-old activity, and, therefore, many possibilities can be envisaged. Possible topics of future work are better compression of higher order resonances; the effect of the strip (stopband), or slot (passband) width; and, in the case of open rings, the positioning of the gap. Some work is already being carried out, including passband filters in multilayer structures, for which the first results should be published soon. Similarly, double-sided FSSs are being implemented. The miniaturization and multi-resonance capabilities of the Matryoshka geometries can be even more effective in 3D configurations, taking advantage of the now affordable 3D printing technology. Moreover, the use of new materials, as well as metamaterials, can offer specific characteristics that need to be explored. With adequate supporting materials, as well as fabrication and measurement facilities, the designs and applications, now limited to the microwave region, can be extended to the millimeter wave and Terahertz frequency bands. Optimized Matryoshka-geometry-based DGS sensors for many specific applications are also something that can be developed in the short term.

We would also like to highlight that the applications developed so far are conceptual and limited to manufacturing processes available on a laboratory scale. Much more can be done from the concepts presented. Moreover, the exploration of the mechanical and electrical properties of new flexible mesh composite materials can enhance the reconfigurability of these structures [81].

To conclude, we highlight that the investigation of the applications of Matryoshka geometries is an open field, whose results obtained so far encourage other groups to get involved in this research effort.

## Figures and Tables

**Figure 1 micromachines-15-00469-f001:**
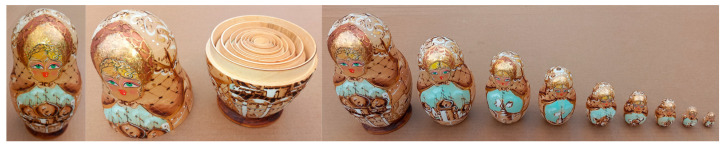
Example of a Russian Matryoshka with nine nested dolls.

**Figure 2 micromachines-15-00469-f002:**
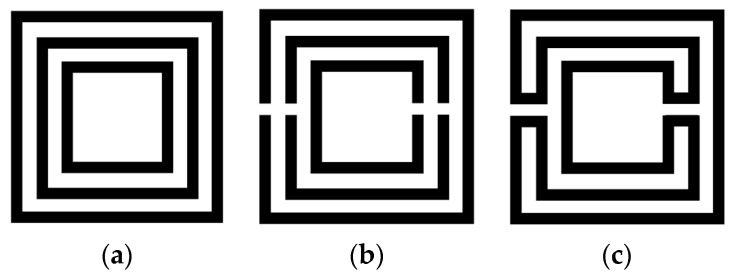
Example of the evolution from concentric rings to a Matryoshka geometry. (**a**) Concentric rings without gaps. (**b**) SRR. (**c**) Matryoshka geometry.

**Figure 3 micromachines-15-00469-f003:**
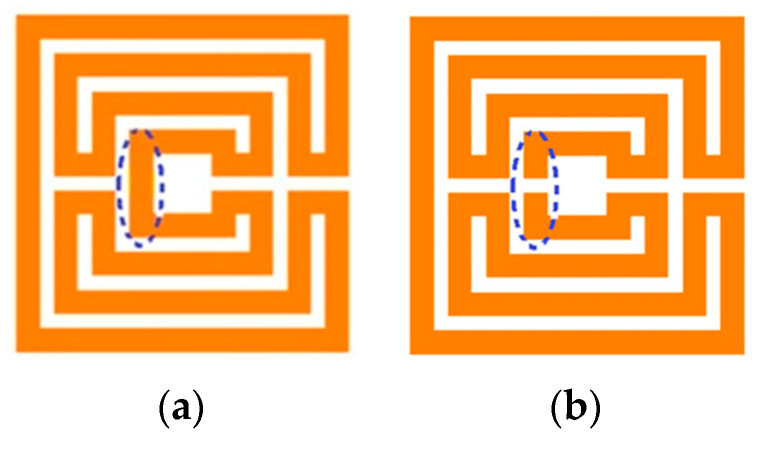
Example of Matryoshka geometries. (**a**) Closed. (**b**) Open.

**Figure 4 micromachines-15-00469-f004:**
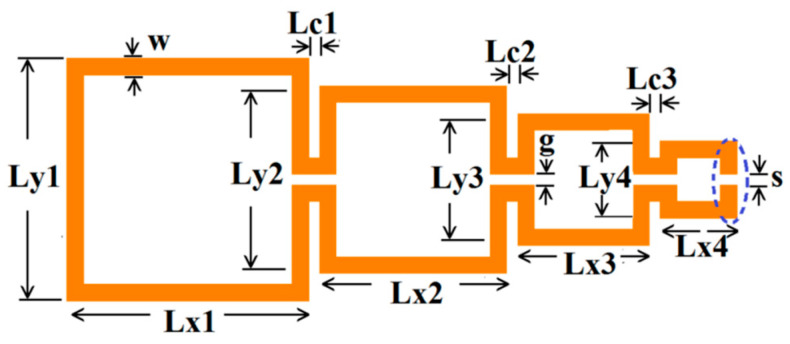
Open Matryoshka geometry, expanded, with the definition of its physical parameters.

**Figure 5 micromachines-15-00469-f005:**
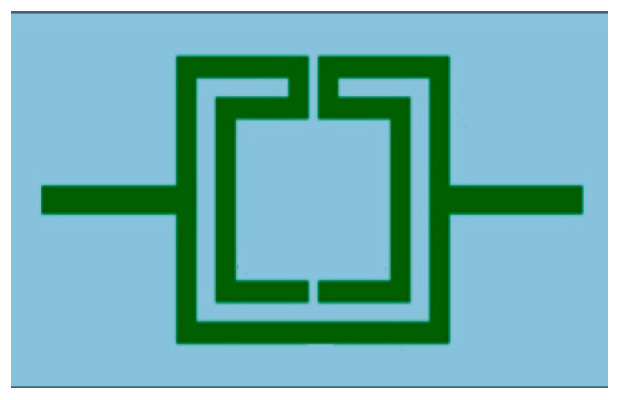
Example of microstrip filter based on an open Matryoshka geometry with two rings.

**Figure 6 micromachines-15-00469-f006:**
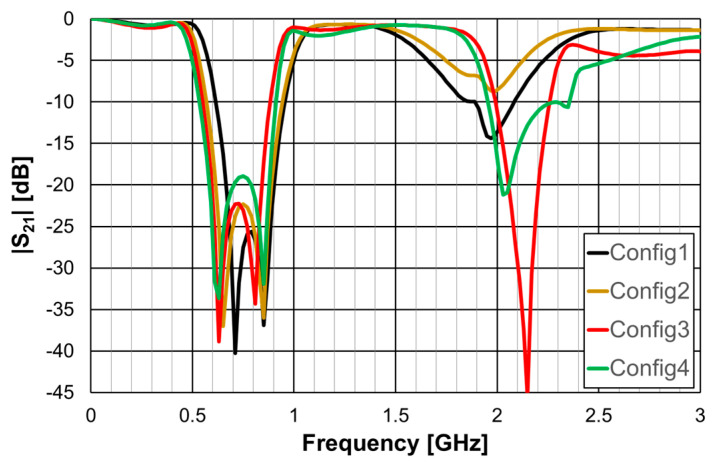
Simulated transmission coefficient of the open Matryoshka filters with constant length.

**Figure 7 micromachines-15-00469-f007:**
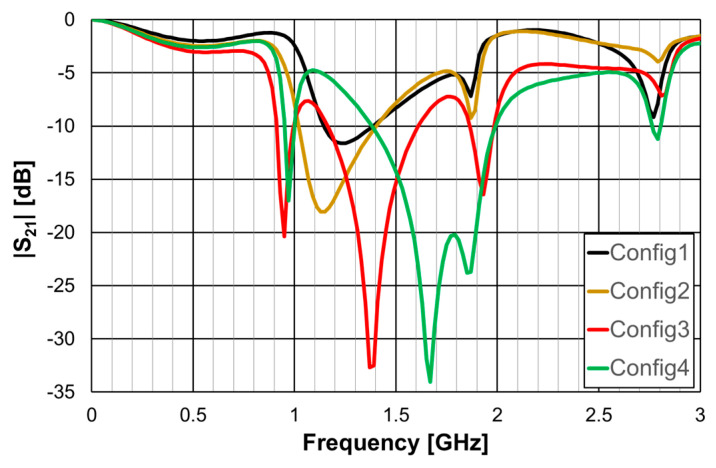
Simulated transmission coefficient of the closed Matryoshka filters with constant length.

**Figure 8 micromachines-15-00469-f008:**
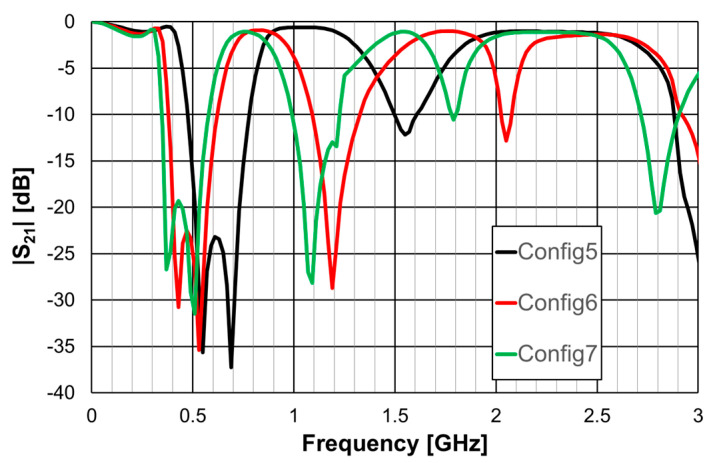
Simulated transmission coefficient of open Matryoshka filters with 2, 3, and 4 rings.

**Figure 9 micromachines-15-00469-f009:**
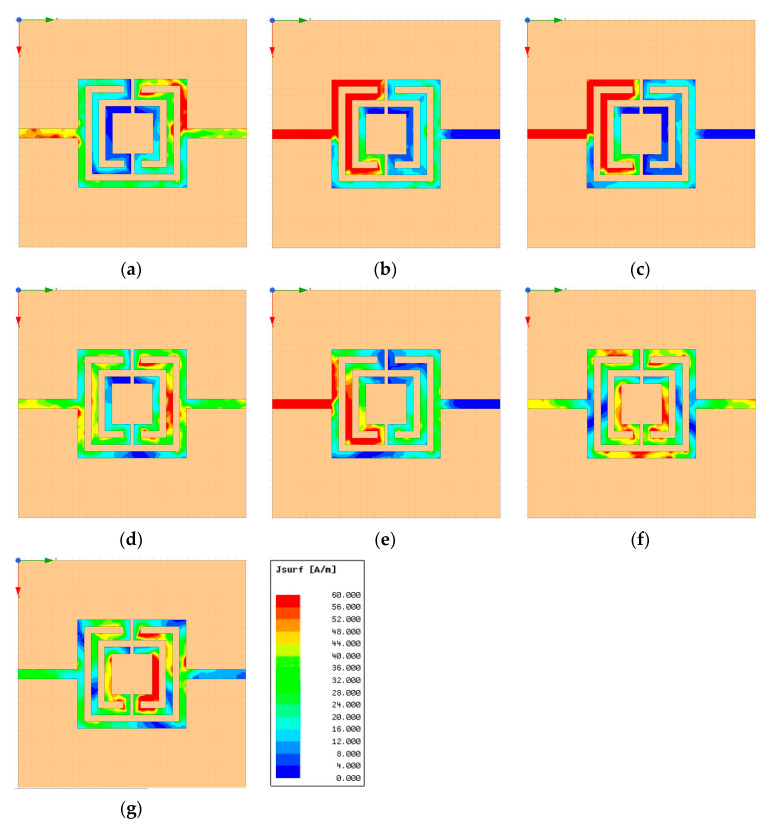
Simulated surface current density of the open Matryoshka filter with three rings (configuration 6). (**a**) f = 0.33 GHz, (**b**) f = f_r1_ = 0.43 GHz, (**c**) f = f_r2_ = 0.53 GHz, (**d**) f = 0.83 GHz, (**e**) f = f_r3_ = 1.19 GHz, (**f**) f = 1.79 GHz, (**g**) f = f_r4_ = 2.05 GHz.

**Figure 10 micromachines-15-00469-f010:**
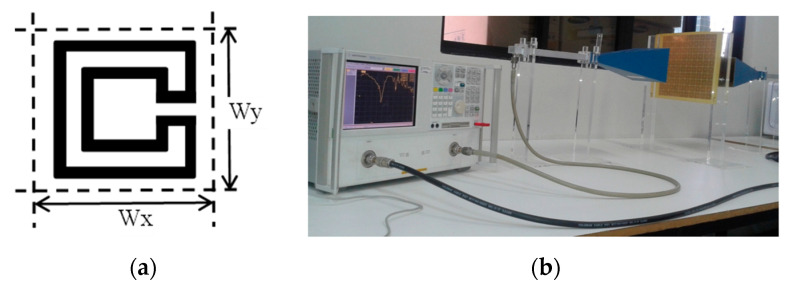
Closed Matryoshka FSS. (**a**) Unit-cell geometry. (**b**) Photo of the prototype with 10 × 10 unit-cells and measurement setup.

**Figure 11 micromachines-15-00469-f011:**
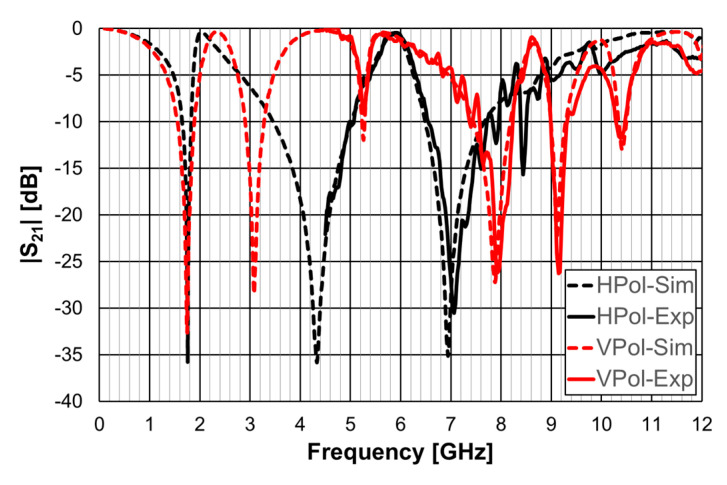
Simulated and experimental |S21| results for configuration 1.

**Figure 12 micromachines-15-00469-f012:**
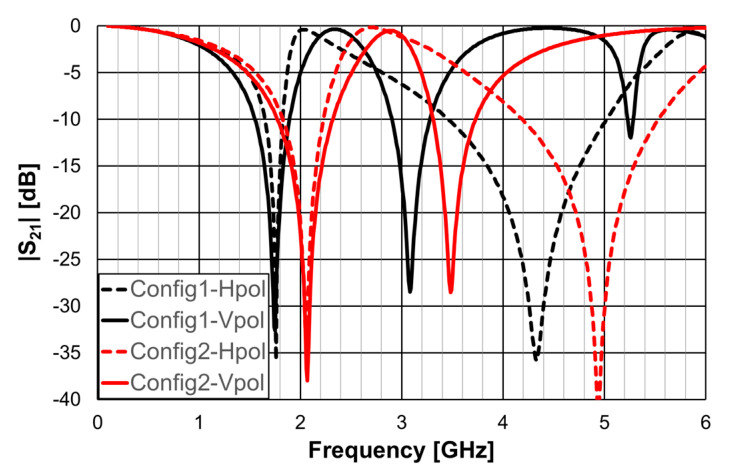
Comparison of the simulated |S21| results for configurations 1 and 2.

**Figure 13 micromachines-15-00469-f013:**
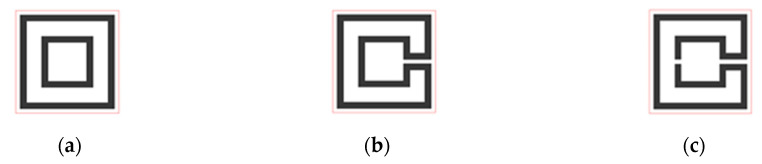
FSS square unit-cells with two square rings. (**a**) Simple rings. (**b**) Closed Matryoshka geometry. (**c**) Open Matryoshka geometry.

**Figure 14 micromachines-15-00469-f014:**
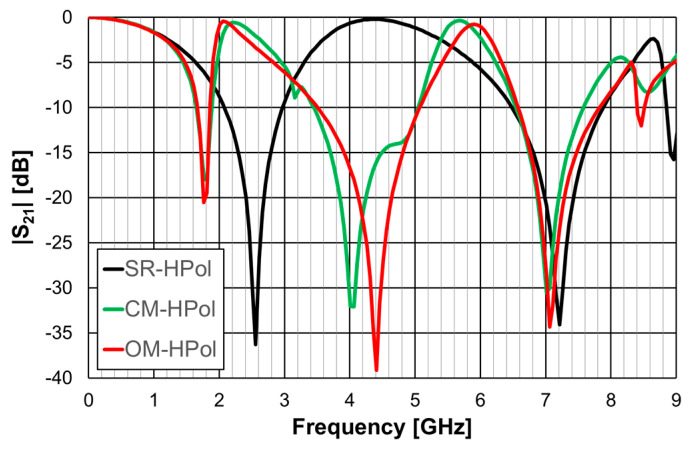
Simulated |S21| results of the simple rings (SR), closed Matryoshka (CM) and open Matryoshka (OM), for horizontal polarization (HPol).

**Figure 15 micromachines-15-00469-f015:**
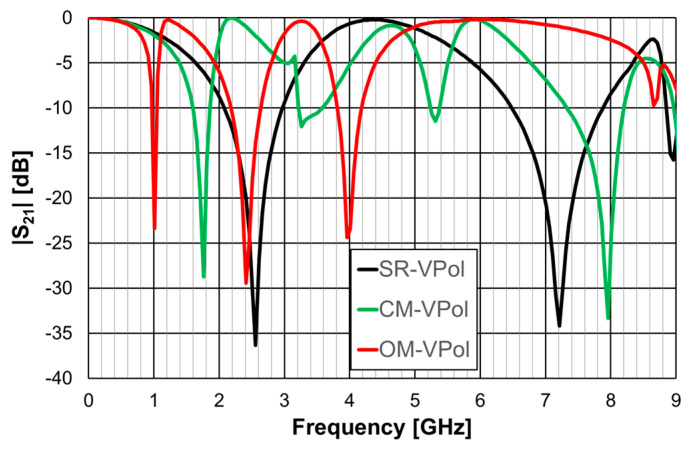
Simulated |S21| results of the simple rings (SR), closed Matryoshka (CM) and open Matryoshka (OM), for vertical polarization (VPol).

**Figure 16 micromachines-15-00469-f016:**
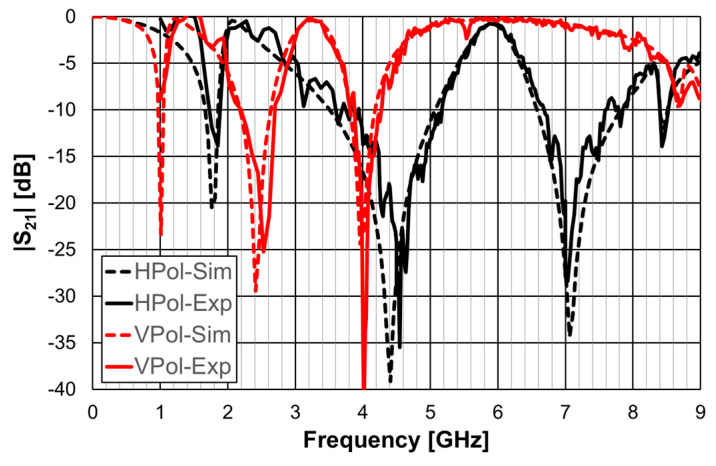
Comparison of simulated and experimental |S21| results of the open Matryoshka for horizontal and vertical polarizations.

**Figure 17 micromachines-15-00469-f017:**
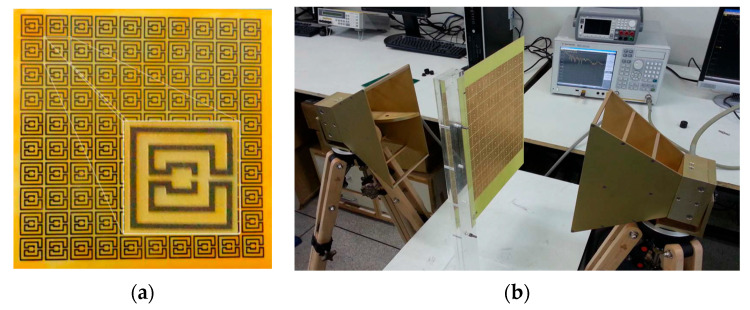
Photo of the FSS test procedure. (**a**) Prototype. (**b**) Experimental setup.

**Figure 18 micromachines-15-00469-f018:**
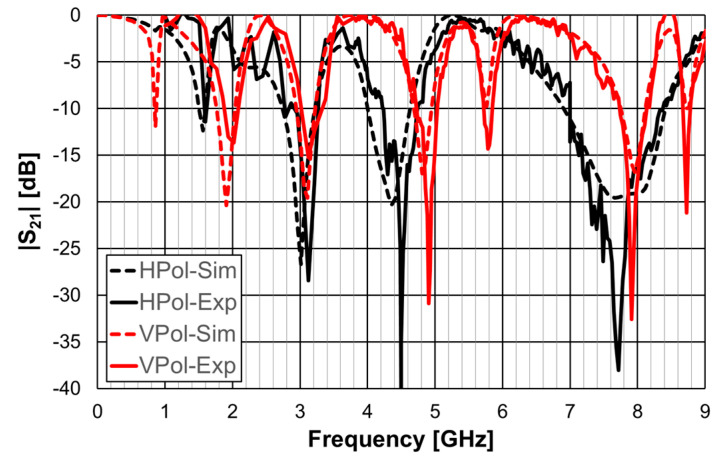
Comparison of simulated and experimental |S21| results of the open Matryoshka with 3 rings for horizontal and vertical polarizations.

**Figure 19 micromachines-15-00469-f019:**
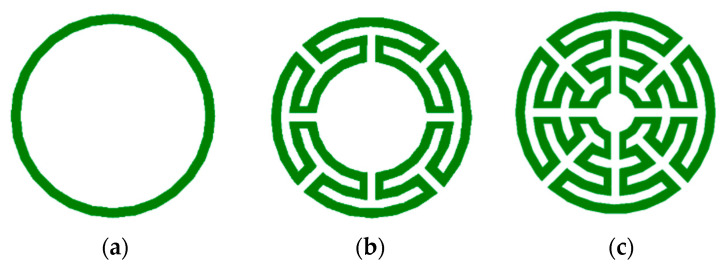
Evolution from a simple circular ring to a circular multiring Matryoshka geometry. (**a**) Simple circular ring. (**b**) Circular Matryoshka geometry with three rings. (**c**) Circular Matryoshka geometry with five rings.

**Figure 20 micromachines-15-00469-f020:**
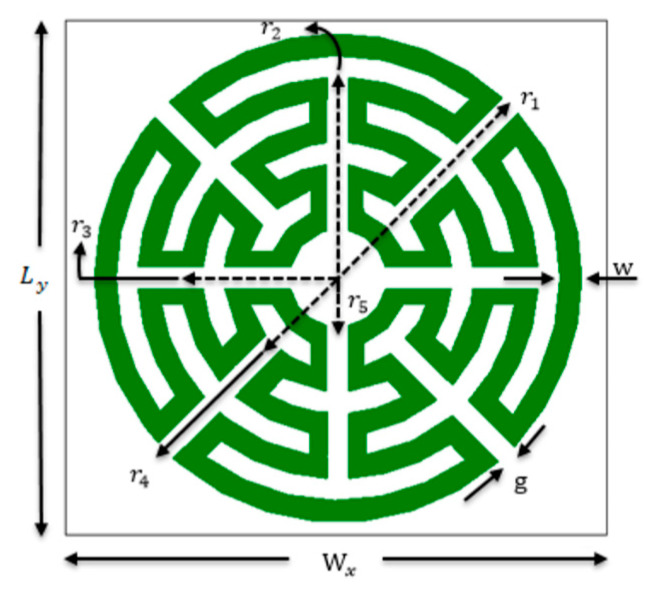
Definition of the physical parameters of an FSS unit-cell with circular multiring Matryoshka geometry.

**Figure 21 micromachines-15-00469-f021:**
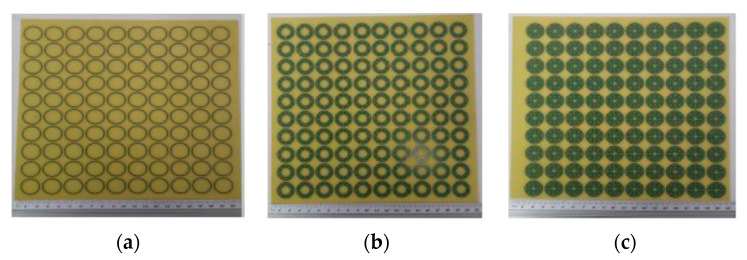
Prototypes of the FSSs with circular Matryoshka unit-cells. (**a**) FSS1; (**b**) FSS2; (**c**) FSS3.

**Figure 22 micromachines-15-00469-f022:**
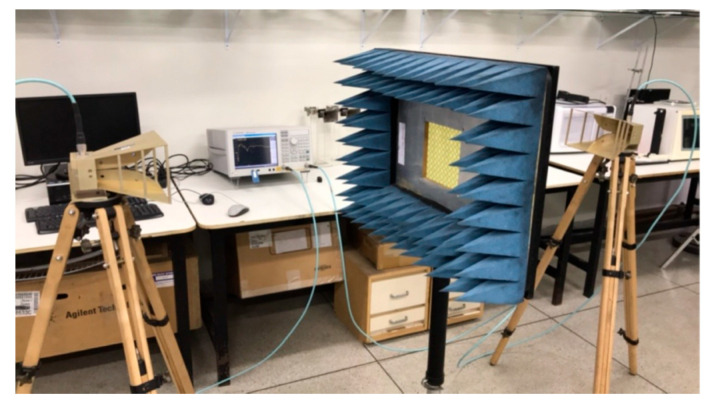
Setup for the measurement of the FSS prototypes with circular Matryoshka unit-cells.

**Figure 23 micromachines-15-00469-f023:**
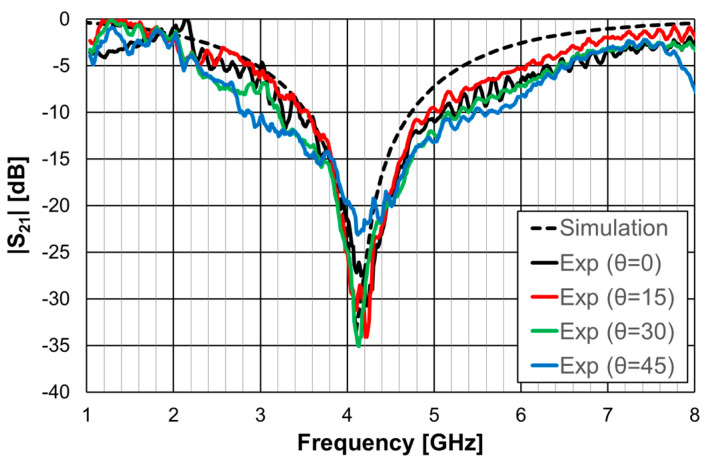
Comparison of the simulated and experimental |S21| results of the FSS1 prototype.

**Figure 24 micromachines-15-00469-f024:**
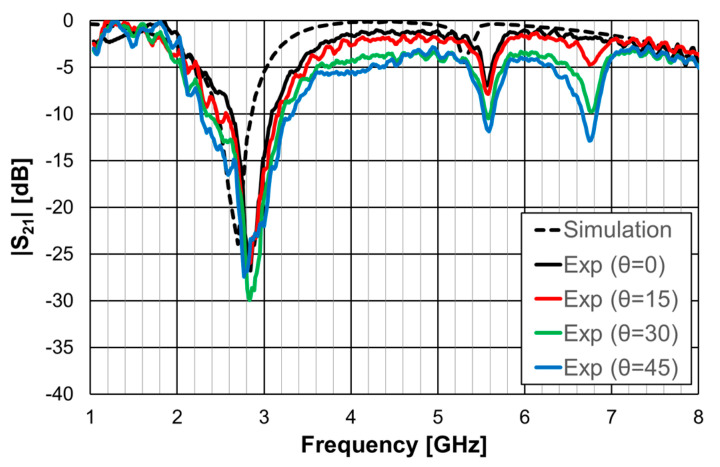
Comparison of the simulated and experimental |S21| results of the FSS2 prototype.

**Figure 25 micromachines-15-00469-f025:**
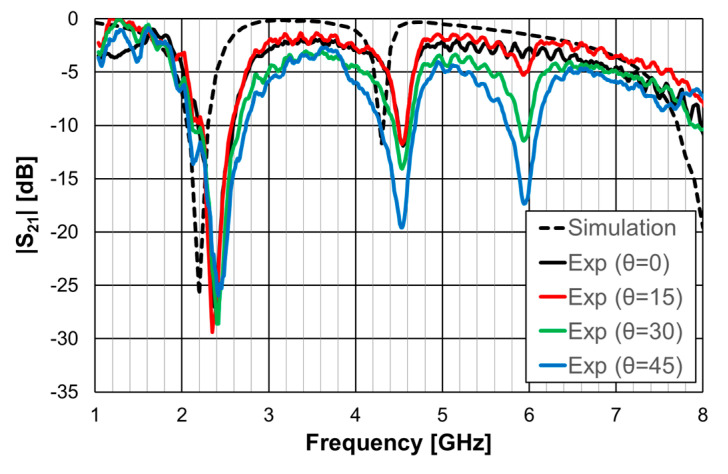
Comparison of the simulated and experimental |S21| results of the FSS3 prototype.

**Figure 26 micromachines-15-00469-f026:**
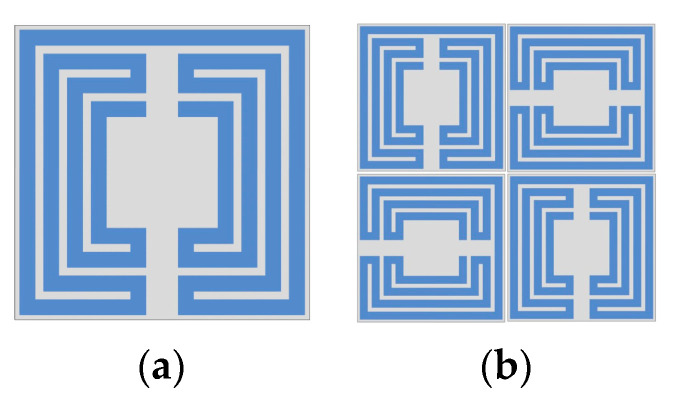
Matryoshka unit-cells proposed in [42]. (**a**) Single element. (**b**) Combination of four orthogonal elements.

**Figure 27 micromachines-15-00469-f027:**
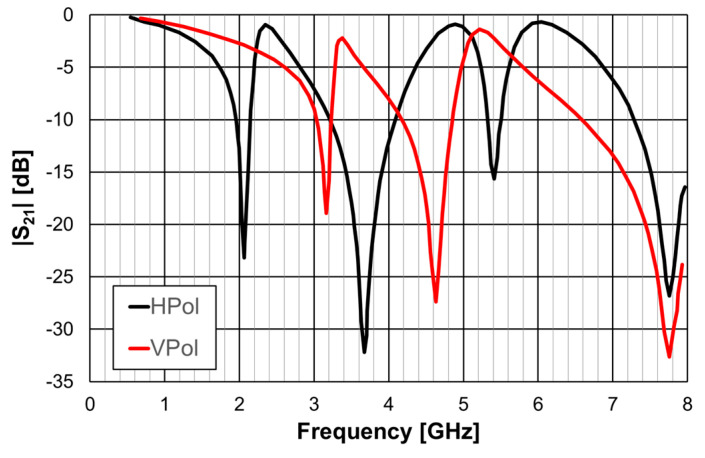
|S21| simulation results of the FSS with the single-element Matryoshka unit-cell.

**Figure 28 micromachines-15-00469-f028:**
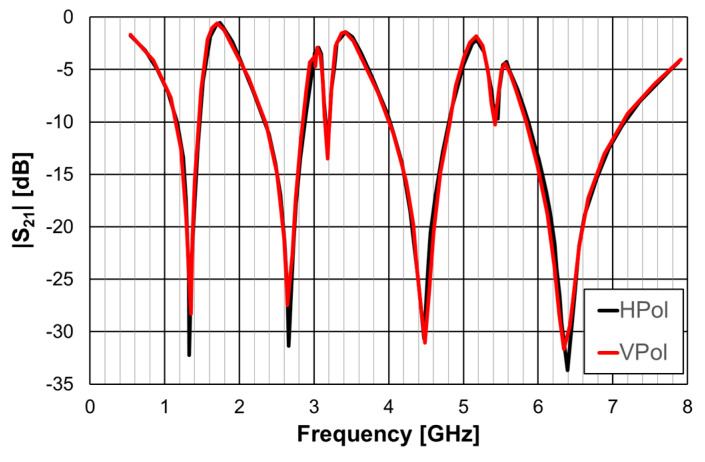
|S21| simulation results of the FSS with the four orthogonal Matryoshka elements unit-cell.

**Figure 29 micromachines-15-00469-f029:**
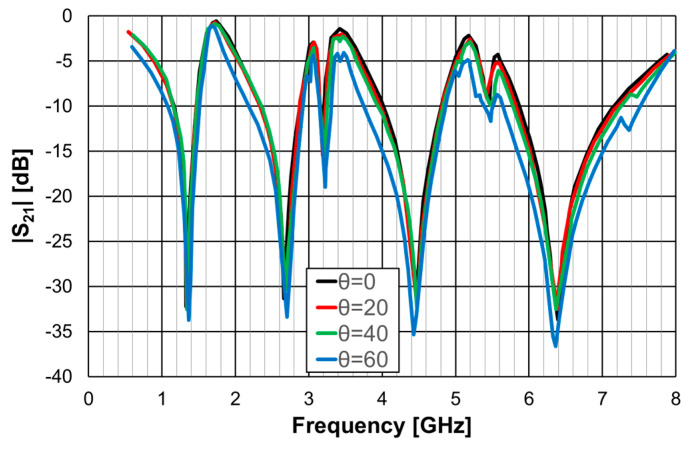
|S21| simulation results of the FSS with the four orthogonal Matryoshka elements unit-cell for horizontal polarization.

**Figure 30 micromachines-15-00469-f030:**
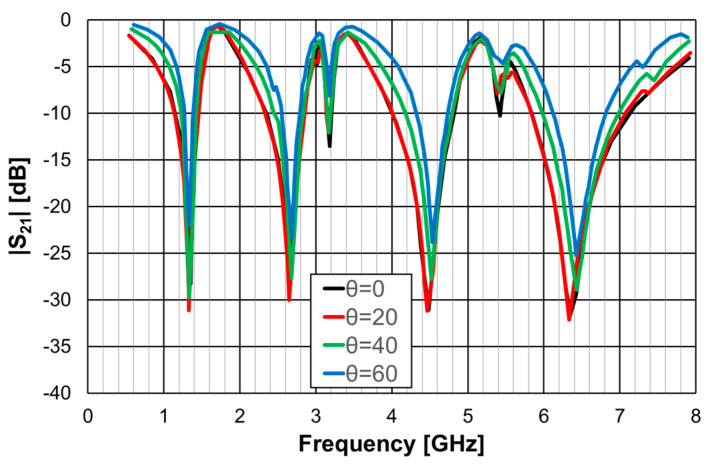
|S21| simulation results of the FSS with the four orthogonal Matryoshka elements unit-cell for vertical polarization.

**Figure 31 micromachines-15-00469-f031:**
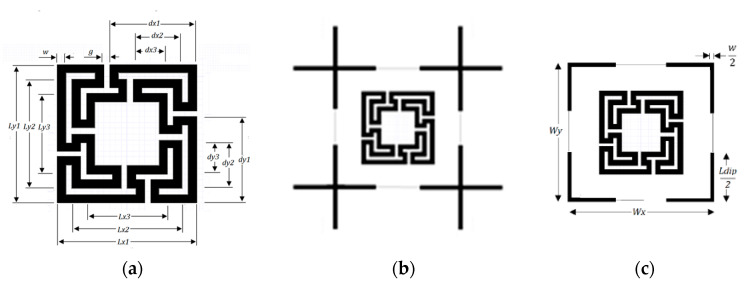
Combination of a Matryoshka geometry with cross-dipoles to form an FSS. (**a**) Matryoshka geometry. (**b**) Combination of Matryoshka with cross-dipoles. (**c**) FSS unit-cell.

**Figure 32 micromachines-15-00469-f032:**
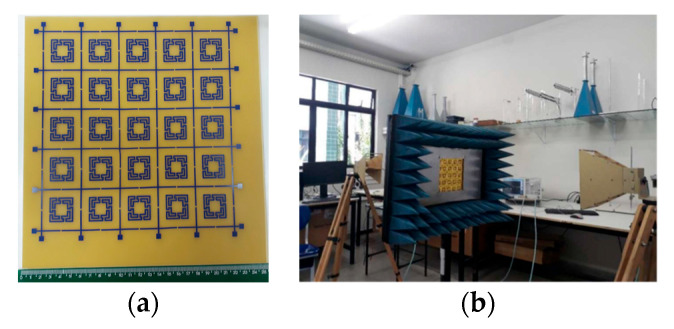
Photos of the FSS with the combination of a Matryoshka geometry with cross-dipoles. (**a**) Prototype. (**b**) Experimental setup.

**Figure 33 micromachines-15-00469-f033:**
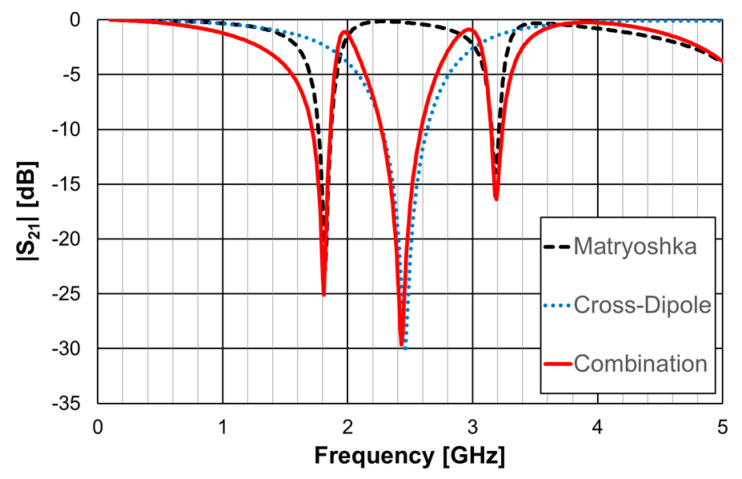
|S21| response of an FSS for the Matryoshka geometry, the cross-dipoles, and the combination of the two.

**Figure 34 micromachines-15-00469-f034:**
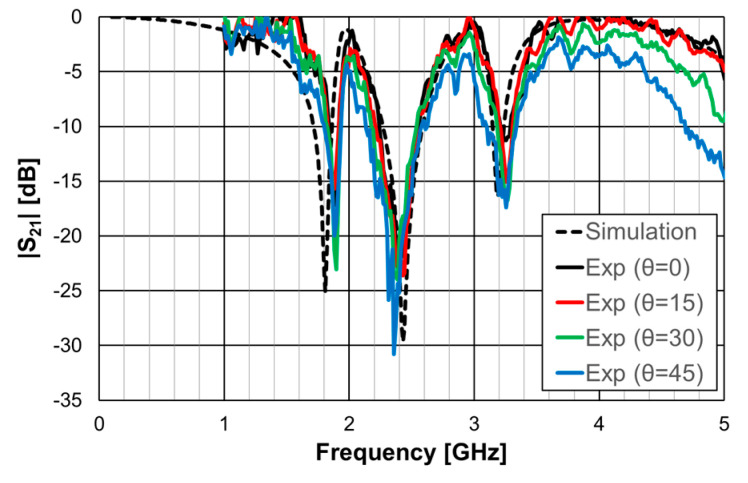
Comparison of the |S21| simulation and experimental results for the FSS combination of the Matryoshka geometry with the cross-dipoles.

**Figure 35 micromachines-15-00469-f035:**
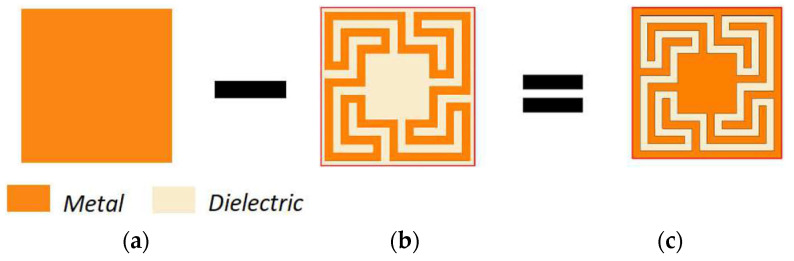
Complementary form of FSS Matryoshka geometry unit-cell. (**a**) Metal patch. (**b**) Metal Matryoshka geometry. (**c**) Complementary Matryoshka geometry.

**Figure 36 micromachines-15-00469-f036:**
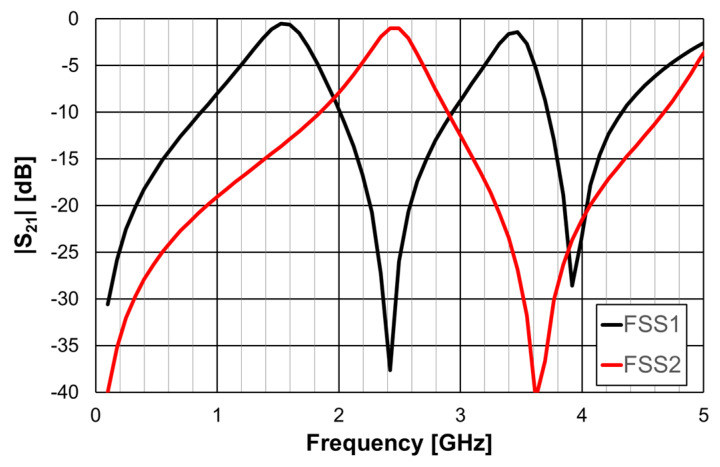
|S21| simulation results of the complimentary Matryoshka configurations FSS1 and FSS2.

**Figure 37 micromachines-15-00469-f037:**
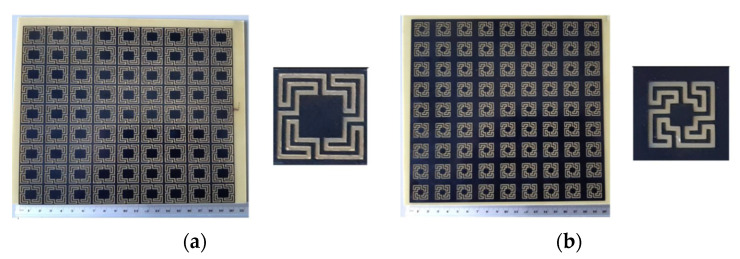
Photos of the prototypes of the complimentary Matryoshka configurations FSS1 and FSS2. (**a**) FSS1. (**b**) FSS2.

**Figure 38 micromachines-15-00469-f038:**
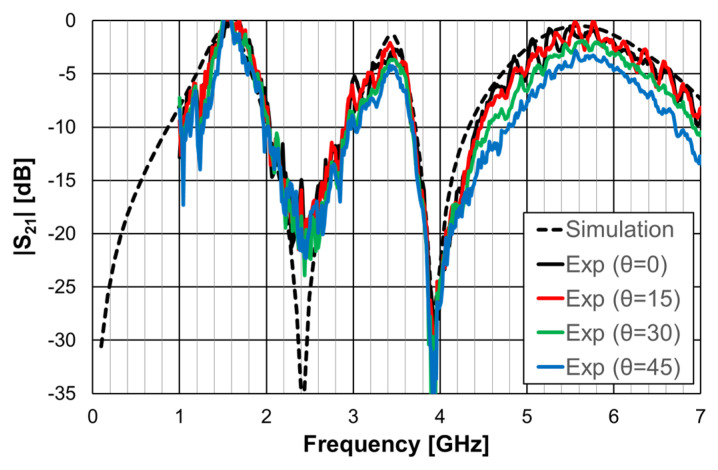
Comparison of the |S21| simulation and experimental results for the FSS1.

**Figure 39 micromachines-15-00469-f039:**
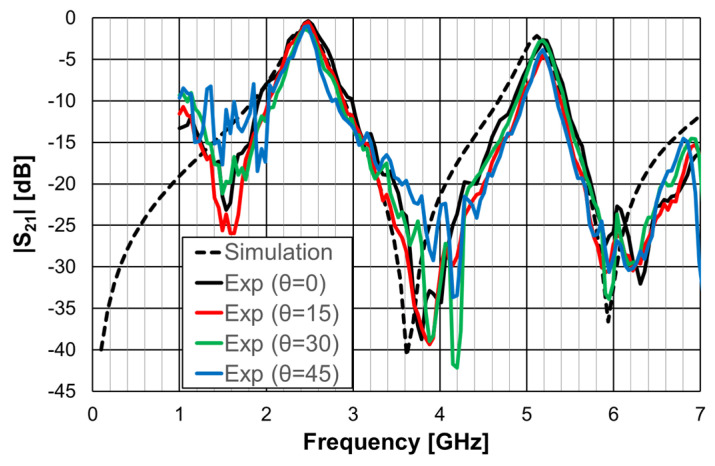
Comparison of the |S21| simulation and experimental results for the FSS2.

**Figure 40 micromachines-15-00469-f040:**

RFSS with Matryoshka geometry. (**a**) Unit-cell. (**b**) 7 × 7 configuration. (**c**) Prototype.

**Figure 41 micromachines-15-00469-f041:**
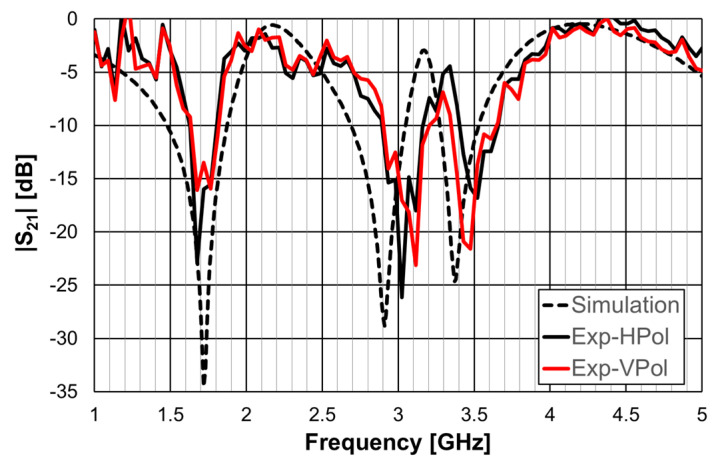
|S21| response of the FSS without PIN diodes and without inductors.

**Figure 42 micromachines-15-00469-f042:**
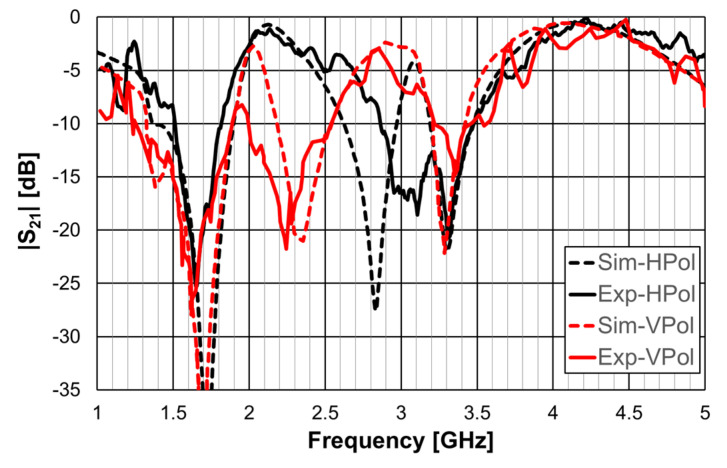
|S21| response of the FSS with PIN diodes but without inductors.

**Figure 43 micromachines-15-00469-f043:**
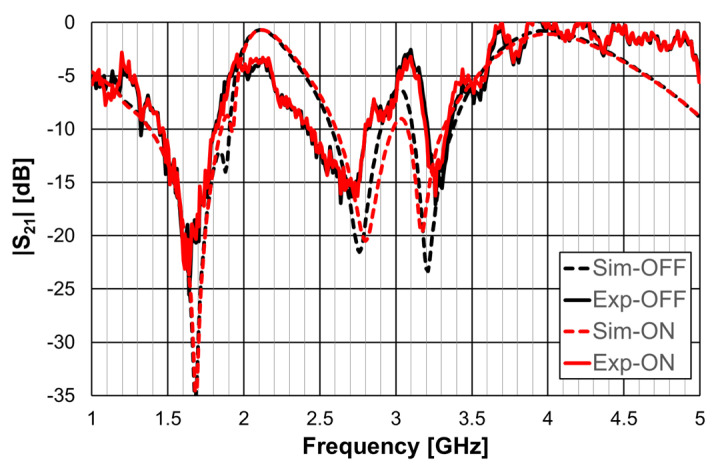
|S21| response of the FSS with PIN diodes and inductors for horizontal polarization.

**Figure 44 micromachines-15-00469-f044:**
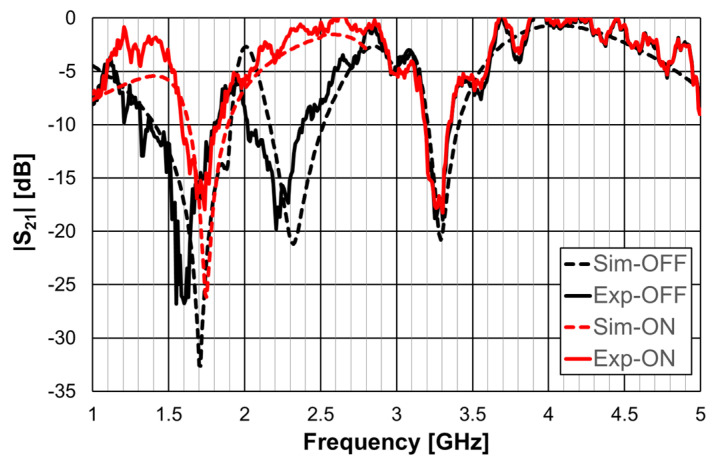
|S21| response of the FSS with PIN diodes and inductors for vertical polarization.

**Figure 45 micromachines-15-00469-f045:**
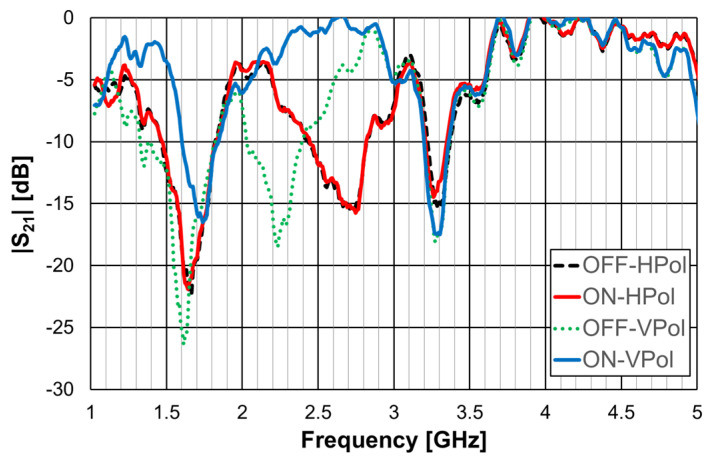
|S21| response of the FSS with PIN diodes and inductors for ON and OFF PIN states.

**Figure 46 micromachines-15-00469-f046:**
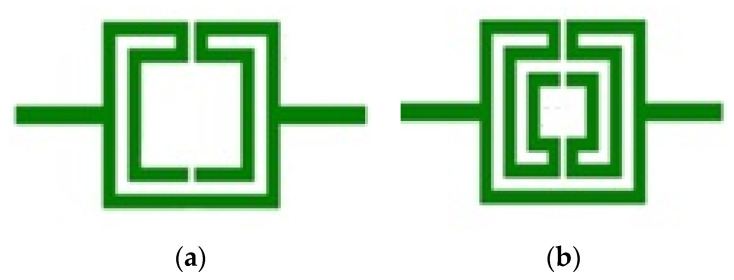
Open Matryoshka square geometry filters. (**a**) With two rings. (**b**) With three rings.

**Figure 47 micromachines-15-00469-f047:**
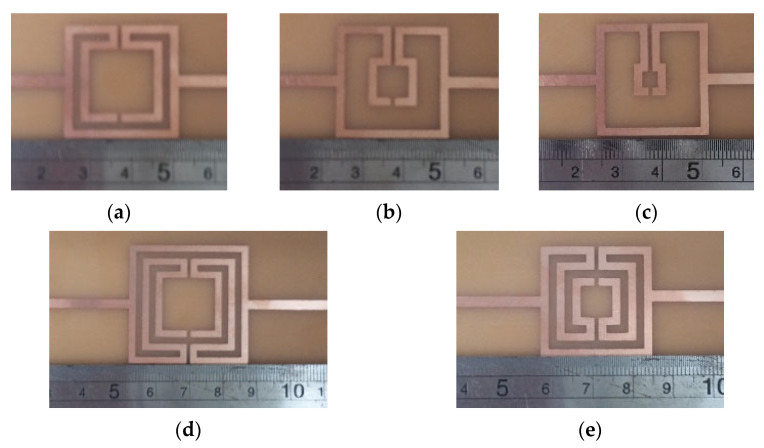
Photos of the prototypes of Matryoshka filters with square rings. (**a**) Config1. (**b**) Config2. (**c**) Config3. (**d**) Config4. (**e**) Config5.

**Figure 48 micromachines-15-00469-f048:**
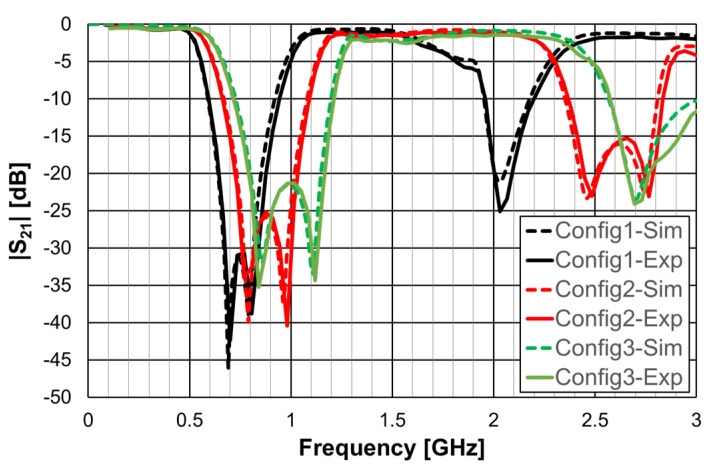
Comparison of the |S21| simulation and experimental results for square Matryoshka filters config1, config2, and config3.

**Figure 49 micromachines-15-00469-f049:**
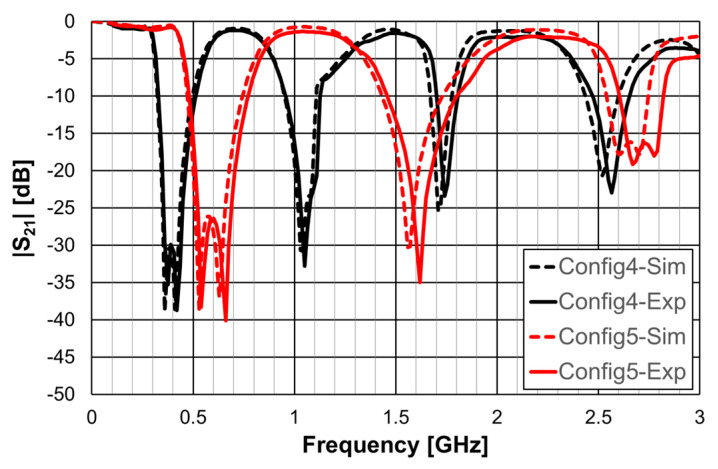
Comparison of the |S21| simulation and experimental results for square Matryoshka filters config4 and config5.

**Figure 50 micromachines-15-00469-f050:**
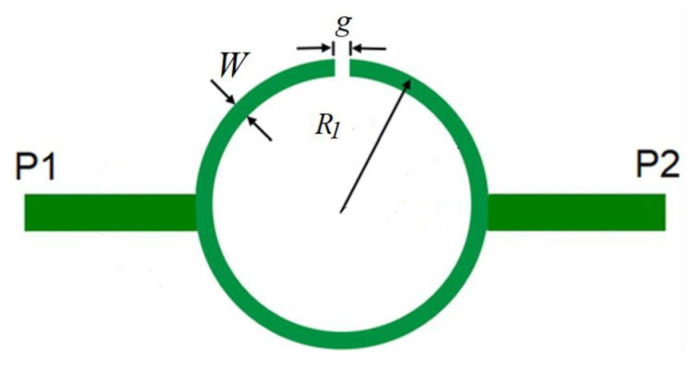
Stopband filter based on an open Matryoshka circular ring geometry.

**Figure 51 micromachines-15-00469-f051:**
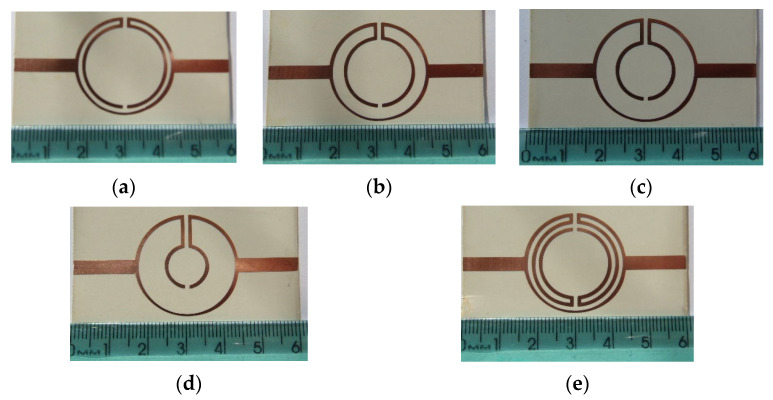
Photos of the prototypes of Matryoshka filters with circular rings. (**a**) Config1. (**b**) Config2. (**c**) Config3. (**d**) Config4. (**e**) Config5.

**Figure 52 micromachines-15-00469-f052:**
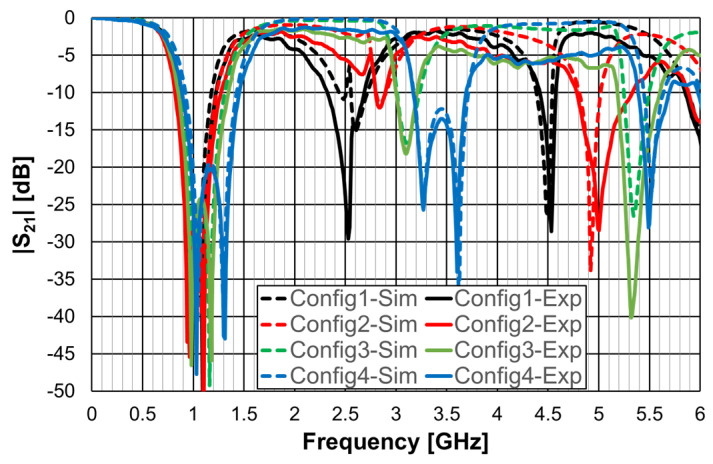
Comparison of the |S21| simulation and experimental results for circular Matryoshka filters config1, config2, config3, and config4.

**Figure 53 micromachines-15-00469-f053:**
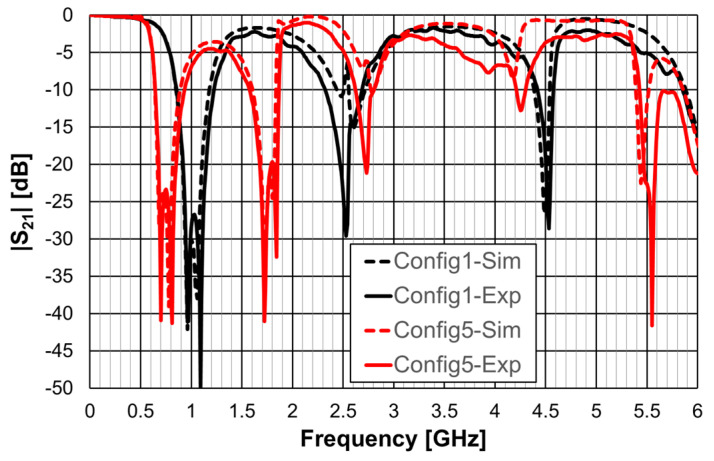
Comparison of the |S21| simulation and experimental results for circular Matryoshka filters config1 and config5.

**Figure 54 micromachines-15-00469-f054:**
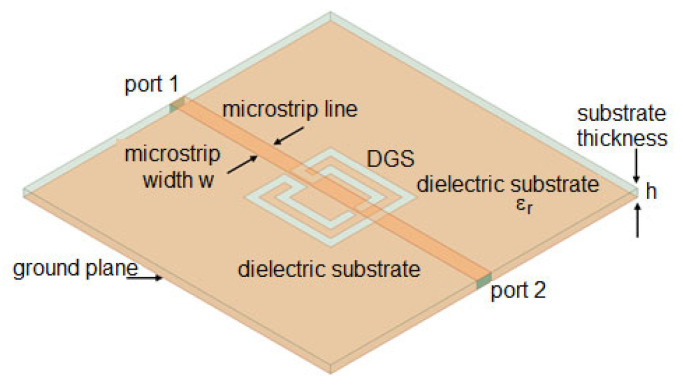
Example of DGS with Matryoshka geometry.

**Figure 55 micromachines-15-00469-f055:**
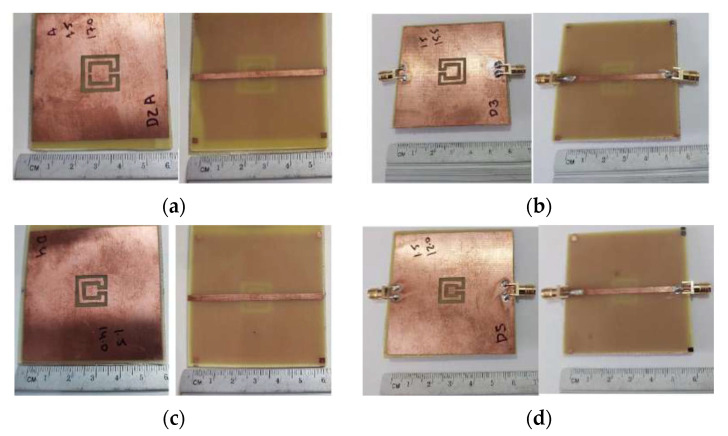
Photographs of the microstrip line with square Matryoshka geometry DGS configurations. (**a**) Config1. (**b**) Config2. (**c**) Config3. (**d**) Config4.

**Figure 56 micromachines-15-00469-f056:**
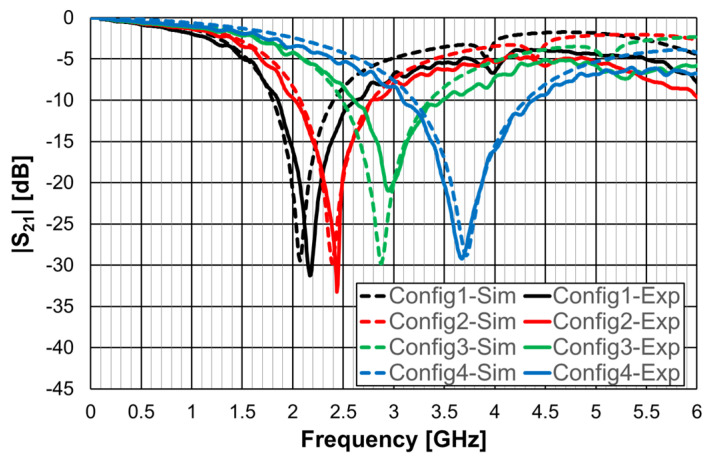
Comparison of the |S21| simulation and experimental responses of a microstrip line with a square Matryoshka geometry DGS.

**Figure 57 micromachines-15-00469-f057:**
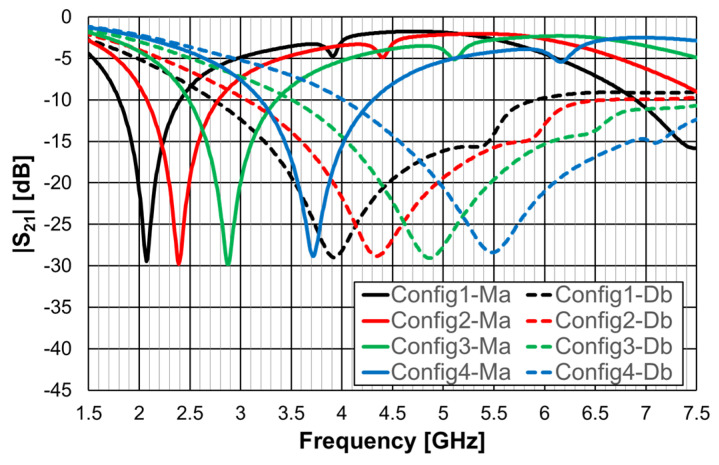
Comparison of the |S21| simulation results of a microstrip line with dumbbell and Matryoshka square geometry DGS.

**Figure 58 micromachines-15-00469-f058:**
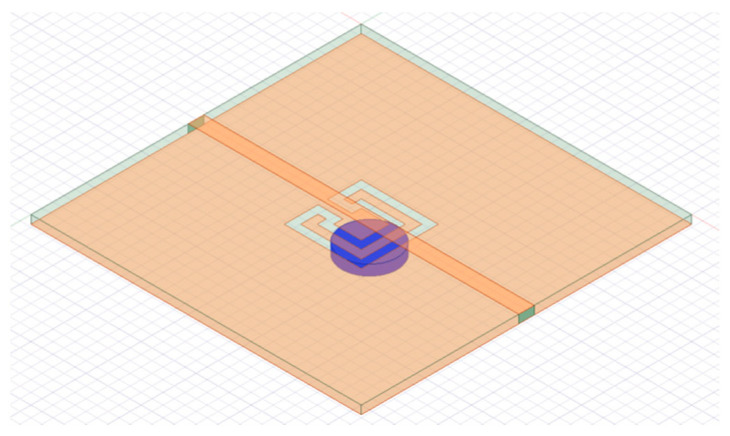
Filter configuration with Matryoshka square geometry DGS and dielectric resonator.

**Figure 59 micromachines-15-00469-f059:**
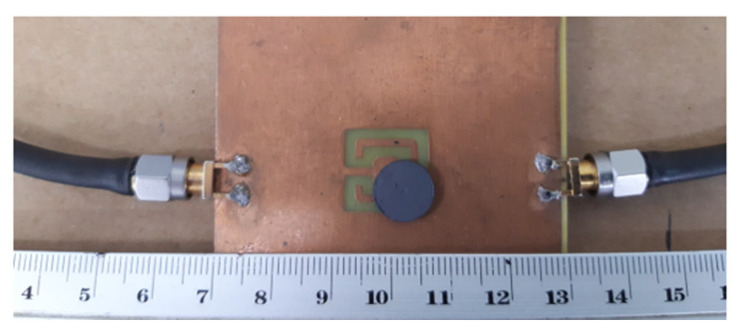
Photo of the prototype of the filter with Matryoshka square geometry DGS and dielectric resonator; bottom view.

**Figure 60 micromachines-15-00469-f060:**
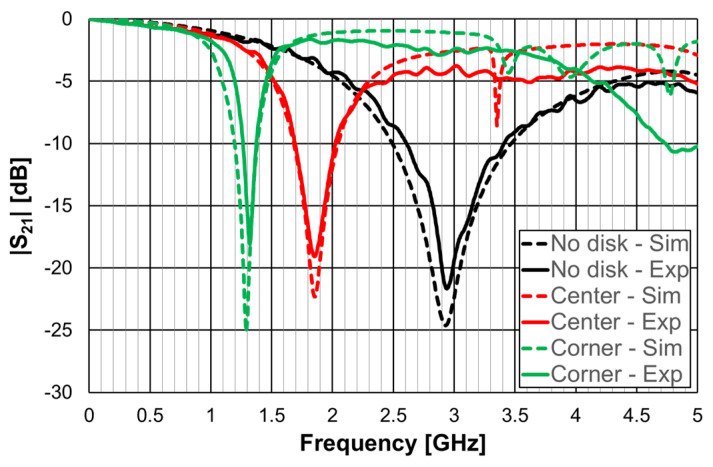
Comparison of the |S21| simulation and experimental results for the square Matryoshka geometry DGS with a dielectric resonator.

**Figure 61 micromachines-15-00469-f061:**
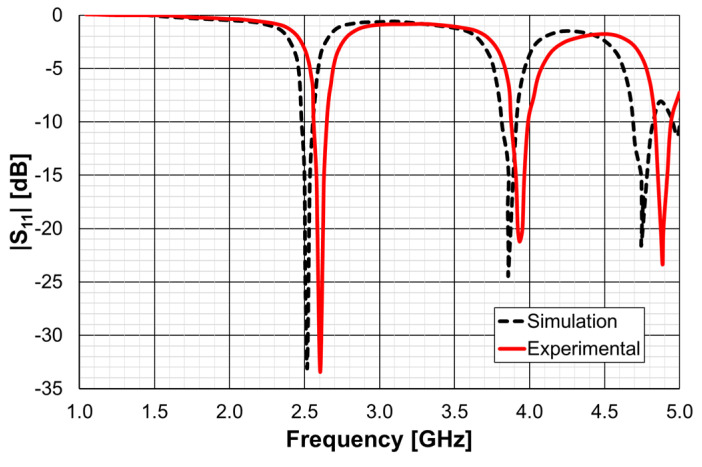
Simulation and experimental input reflection coefficient of the simple rectangular patch.

**Figure 62 micromachines-15-00469-f062:**
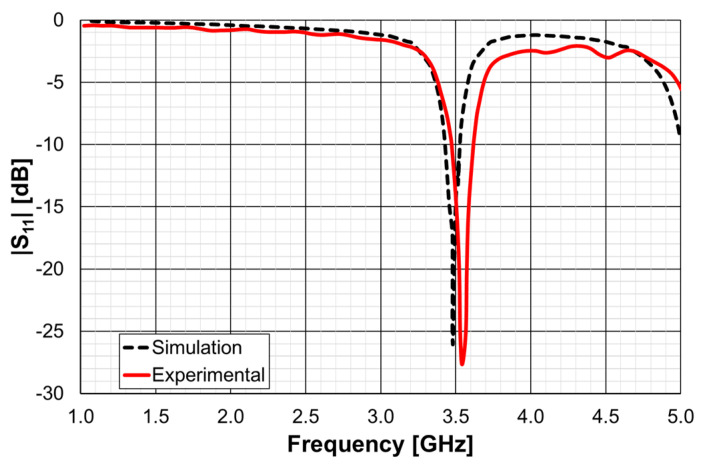
Simulation and experimental amplitude of the input reflection coefficient of the simple rectangular patch.

**Figure 63 micromachines-15-00469-f063:**
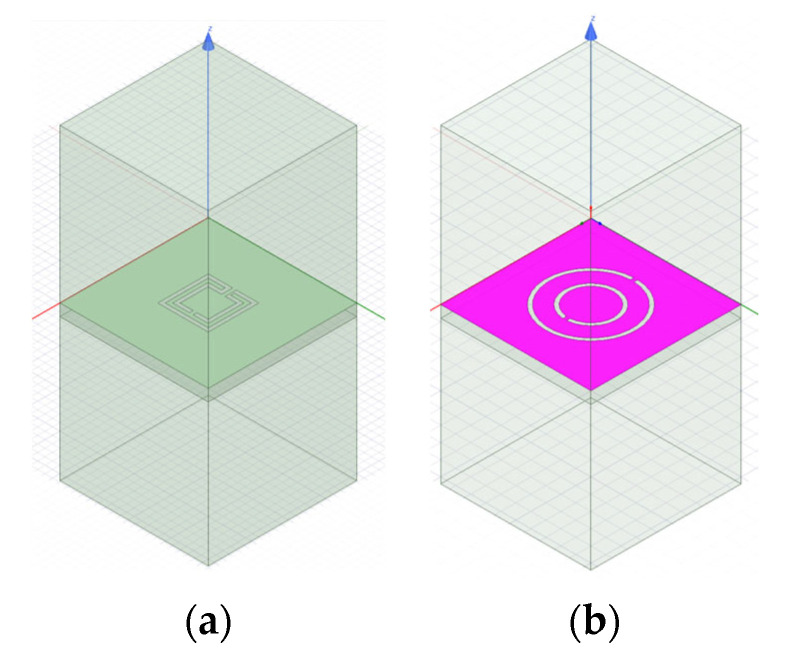
FSS unit-cells. (**a**) Complementary open Matryoshka geometry. (**b**) Complementary circular SRR geometry.

**Figure 64 micromachines-15-00469-f064:**
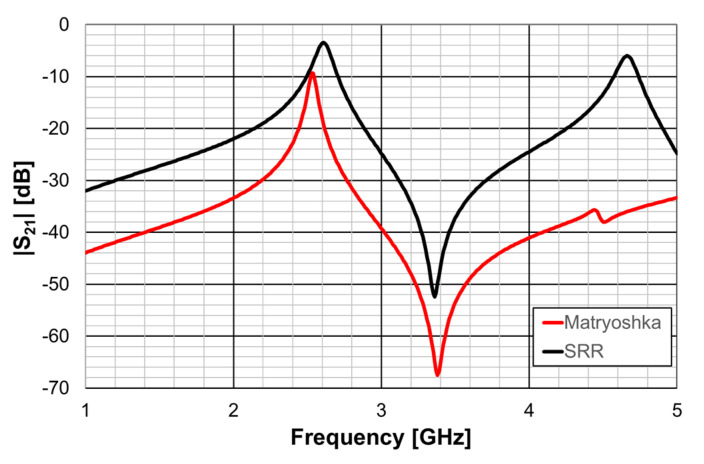
Simulated |S21| results of the open square Matryoshka and SRR FSS configurations.

**Figure 65 micromachines-15-00469-f065:**
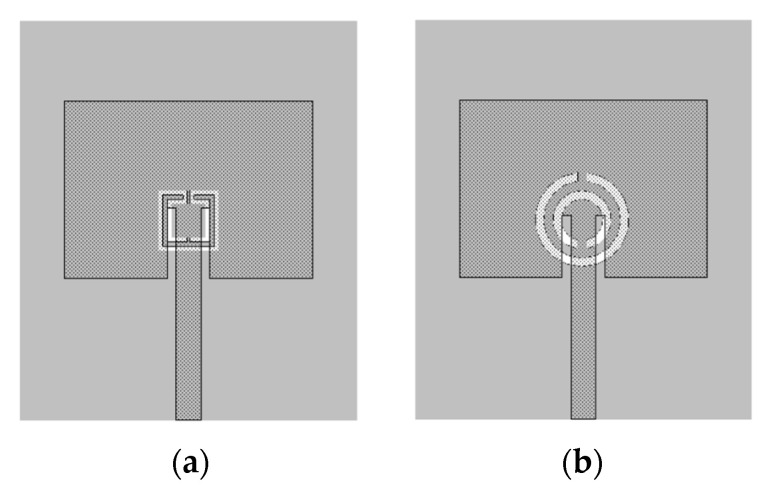
Microstrip patch with DGS. (**a**) Square Matryoshka geometry. (**b**) Circular SRR geometry.

**Figure 66 micromachines-15-00469-f066:**
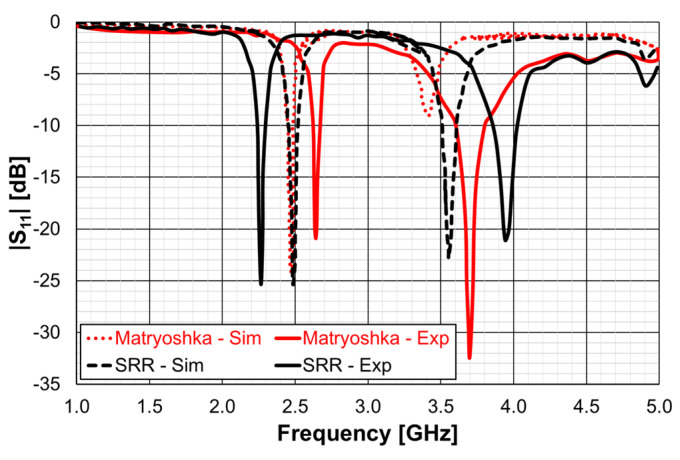
Simulation and experimental |S_11_| results of the patch with DGS ground plane.

**Figure 67 micromachines-15-00469-f067:**
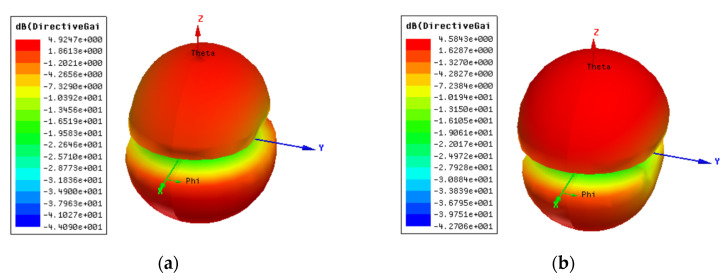
Simulation 3D radiation pattern (gain scale) of the patch with DGS at the first resonance frequency. (**a**) Matryoshka geometry. (**b**) SRR geometry.

**Figure 68 micromachines-15-00469-f068:**
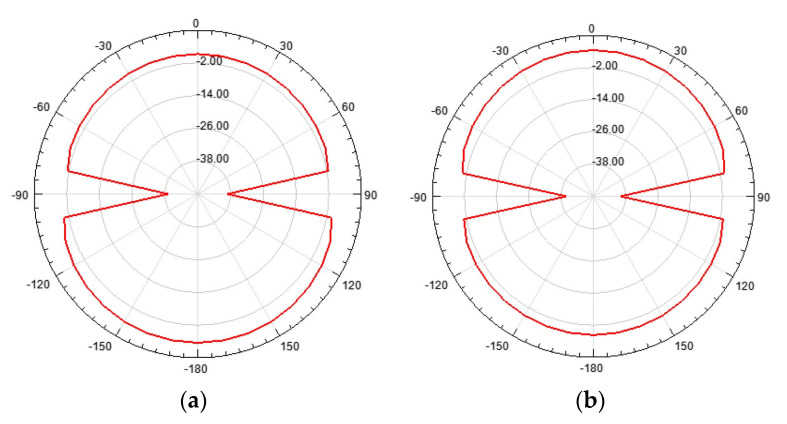
Simulation H-plane radiation pattern (gain scale) of the patch with DGS at the first resonance frequency. (**a**) Matryoshka geometry. (**b**) SRR geometry.

**Figure 69 micromachines-15-00469-f069:**
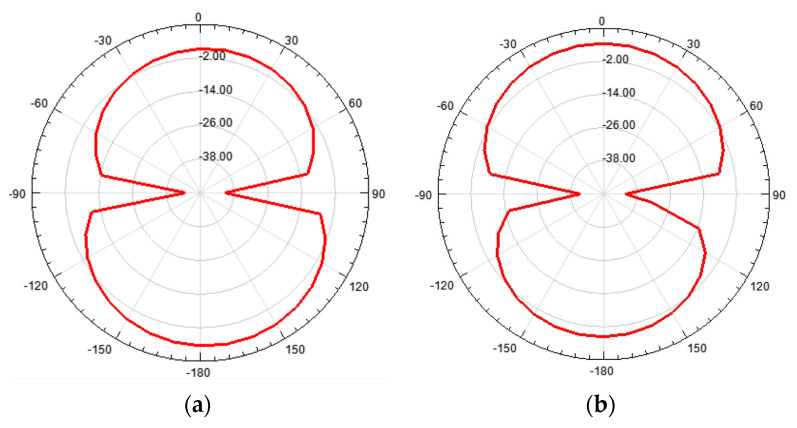
Simulation E-plane radiation pattern (gain scale) of the patch with DGS at the first resonance frequency. (**a**) Matryoshka geometry. (**b**) SRR geometry.

**Figure 70 micromachines-15-00469-f070:**
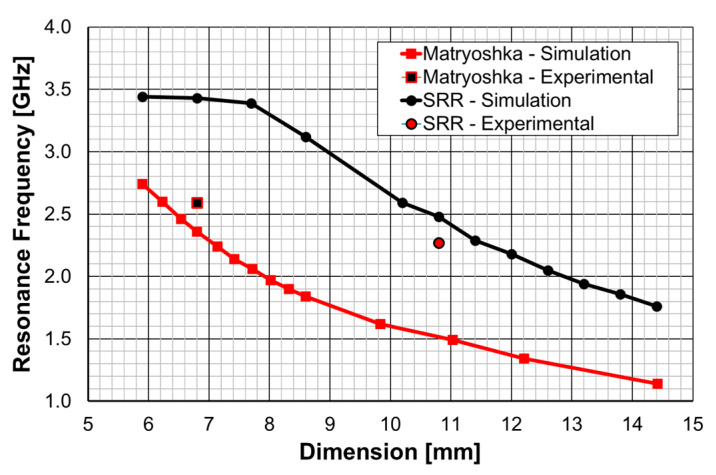
Simulation results for the first resonance frequency of the DGS unit-cell.

**Figure 71 micromachines-15-00469-f071:**
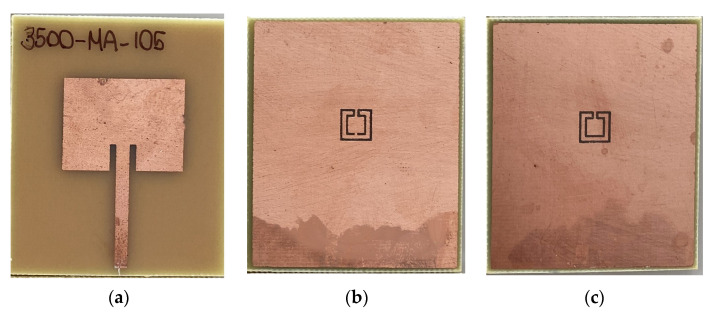
Microstrip patch antenna prototypes with DGS complementary Matryoshka cells. (**a**) Patch side. (**b**) Ground-plane side with an open Matryoshka geometry cell. (**c**) Ground-plane side with a closed Matryoshka geometry cell.

**Figure 72 micromachines-15-00469-f072:**
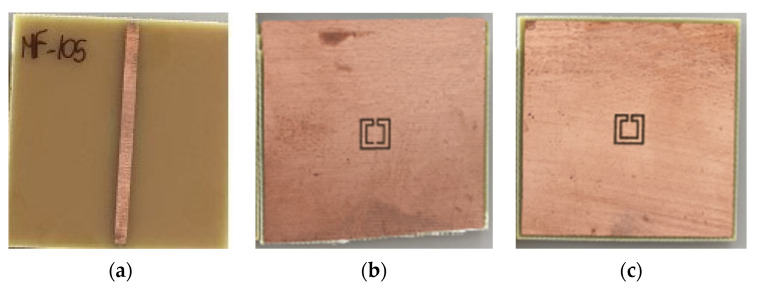
Microstrip line prototypes with DGS complementary Matryoshka cells. (**a**) Microstrip line side. (**b**) Ground-plane side with an open Matryoshka geometry cell. (**c**) Ground-plane side with a closed Matryoshka geometry cell.

**Figure 73 micromachines-15-00469-f073:**
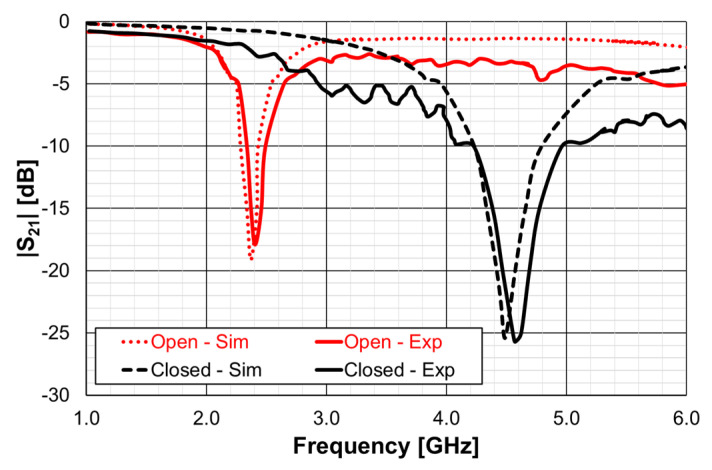
|S21| results for the microstrip line with a DGS with open and closed square Matryoshka cells.

**Figure 74 micromachines-15-00469-f074:**
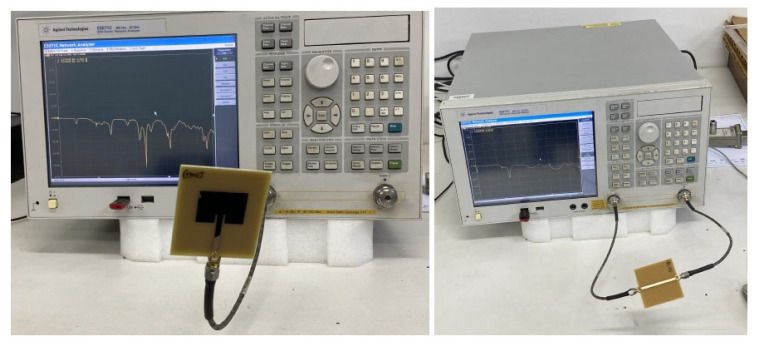
Setups used for the experimental characterization of the microstrip patch and microstrip line with a DGS with open and closed square Matryoshka cells.

**Figure 75 micromachines-15-00469-f075:**
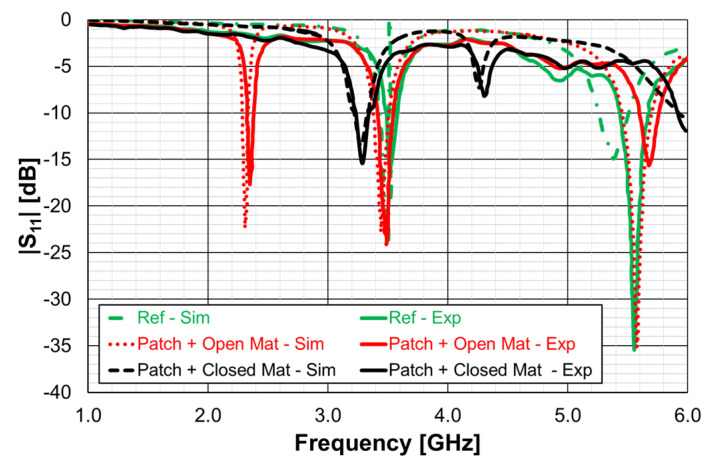
Amplitude of the input reflection coefficient of the patch without and with the DGS ground plane.

**Figure 76 micromachines-15-00469-f076:**
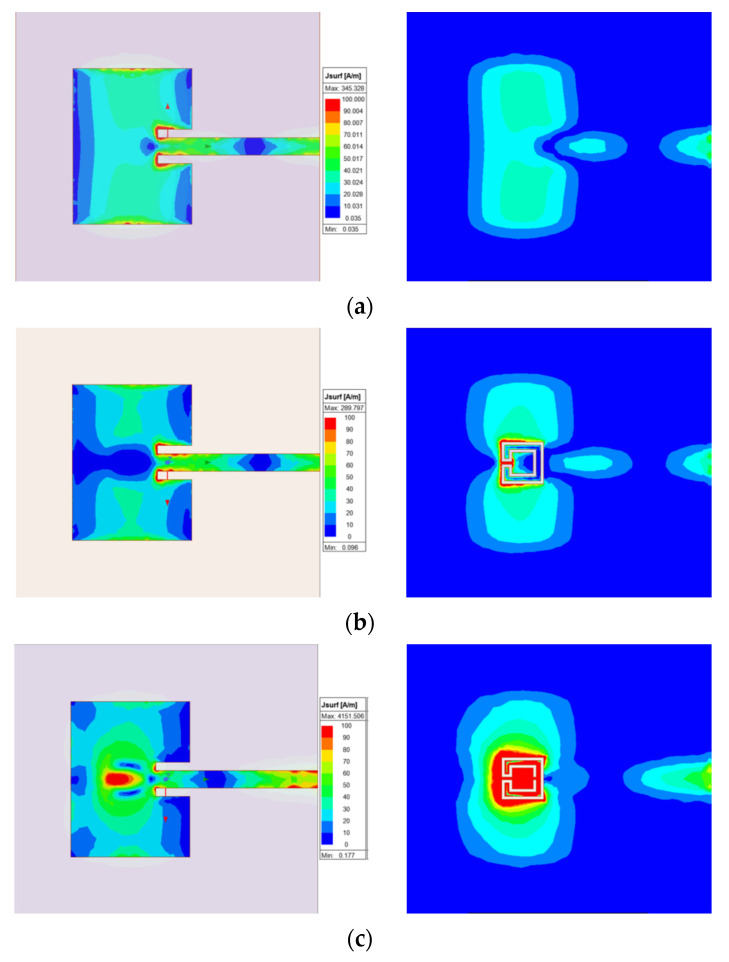
Current distribution on the microstrip patch and ground plane at the first resonance frequency. (**a**) Common patch. (**b**) Patch with closed Matryoshka DGS. (**c**) Patch with open Matryoshka DGS.

**Figure 77 micromachines-15-00469-f077:**
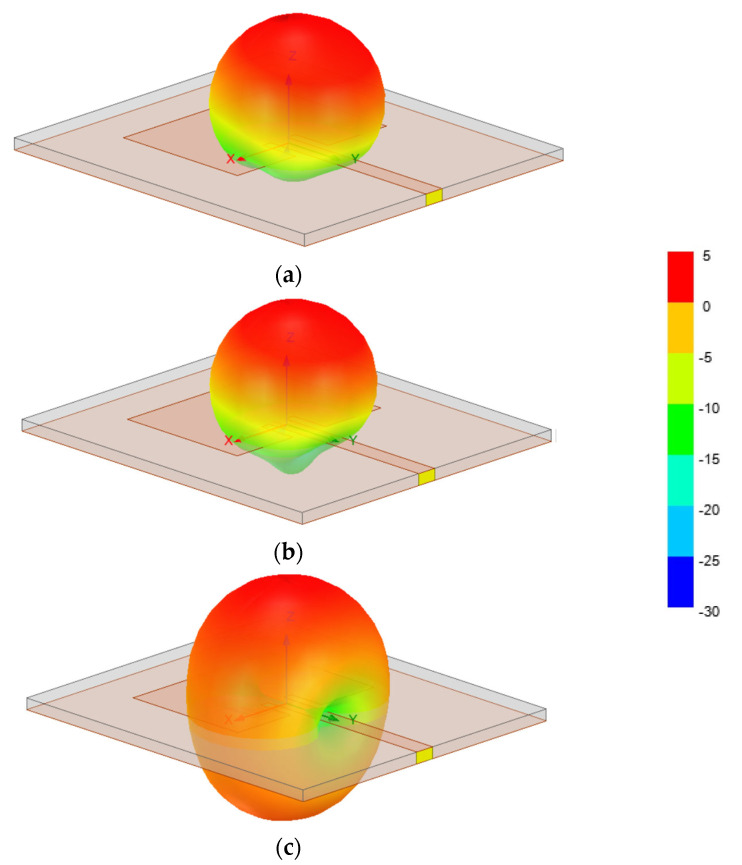
Three-dimensional radiation pattern (gain scale) of the microstrip patch at the first resonance frequency. (**a**) Common patch. (**b**) Patch with closed Matryoshka DGS. (**c**) Patch with open Matryoshka DGS.

**Figure 78 micromachines-15-00469-f078:**
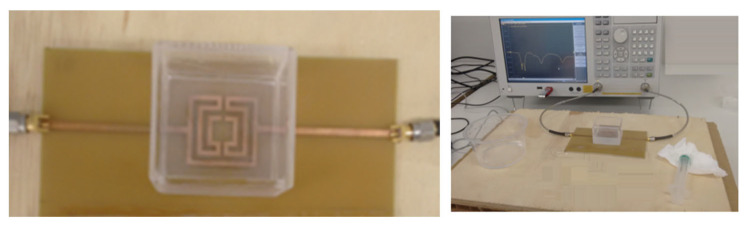
Prototype of the alcohol concentration sensor and of the measurement setup.

**Figure 79 micromachines-15-00469-f079:**
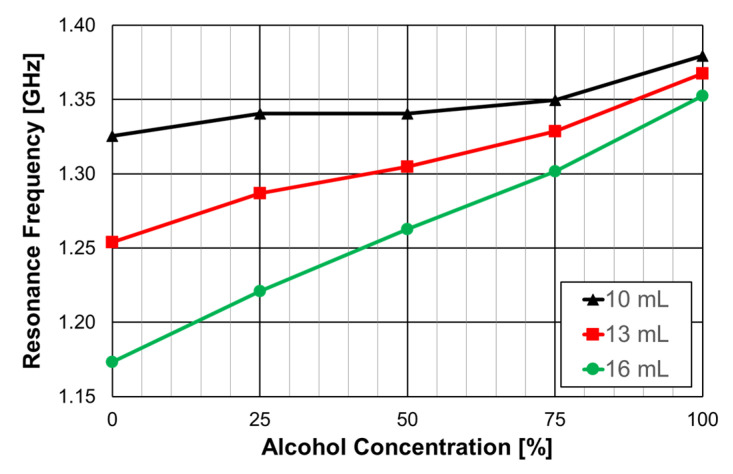
Resonance frequency for different alcohol concentrations and volumes.

**Figure 80 micromachines-15-00469-f080:**
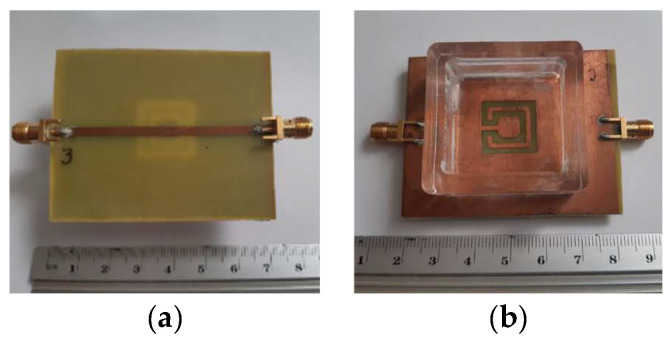
Prototype of the closed Matryoshka geometry DGS sensor. (**a**) Bottom view. (**b**) Top view with container.

**Figure 81 micromachines-15-00469-f081:**
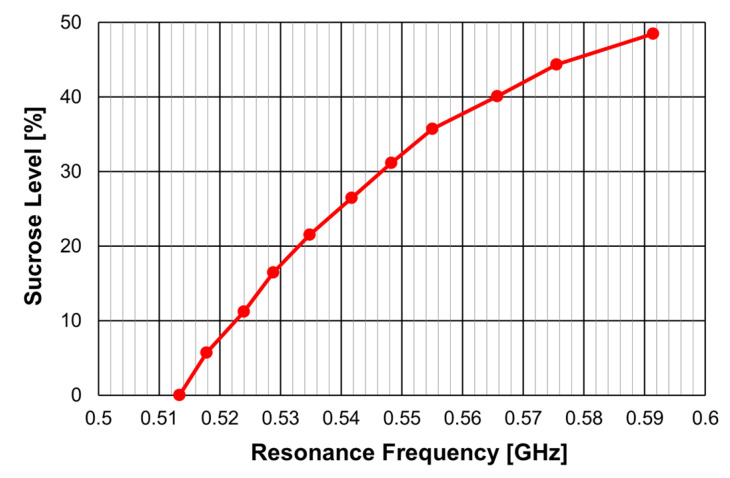
Calibration curve for a sucrose level content sensor.

**Figure 82 micromachines-15-00469-f082:**
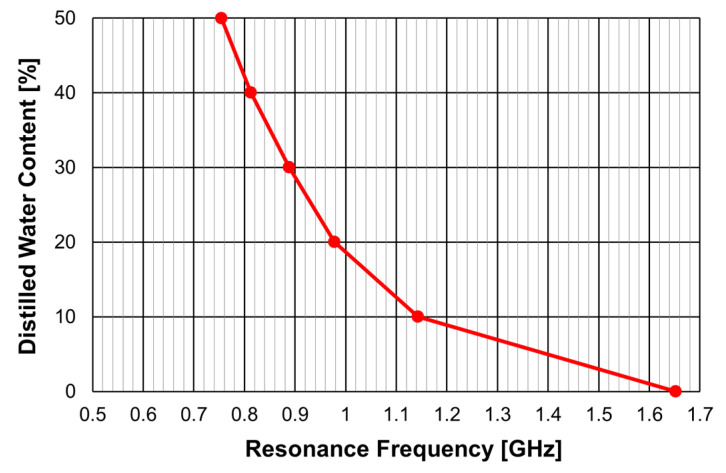
Calibration curve for distilled water content sensor.

**Figure 83 micromachines-15-00469-f083:**
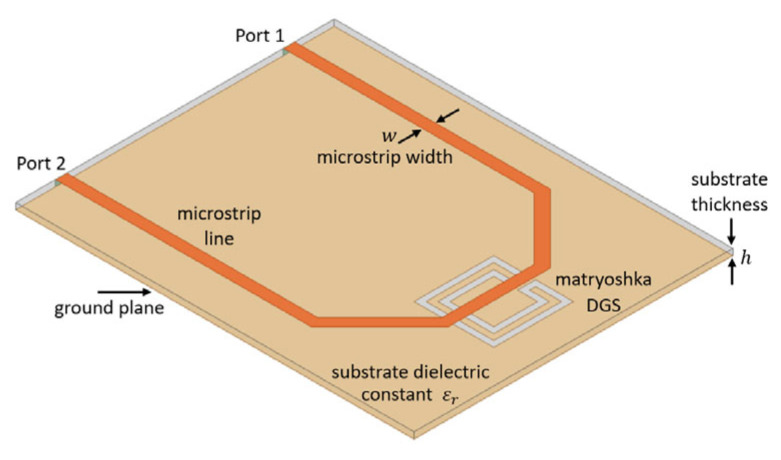
Structure of the filter configuration used as a soil moisture sensor.

**Figure 84 micromachines-15-00469-f084:**
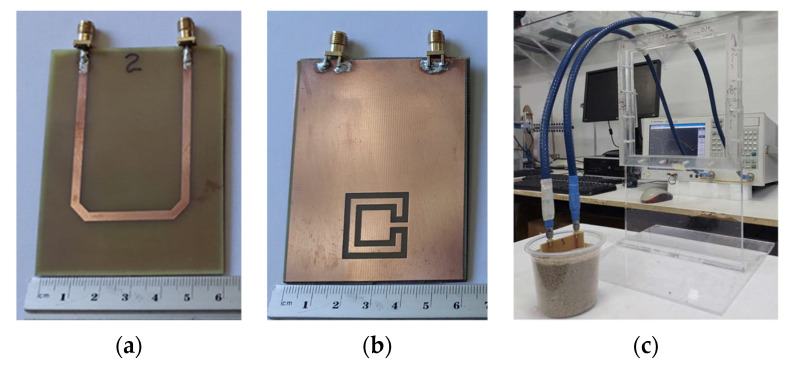
Photos of the experimental validation process. (**a**) Microstrip line side view. (**b**) DGS side view. (**c**) Measurement setup.

**Figure 85 micromachines-15-00469-f085:**
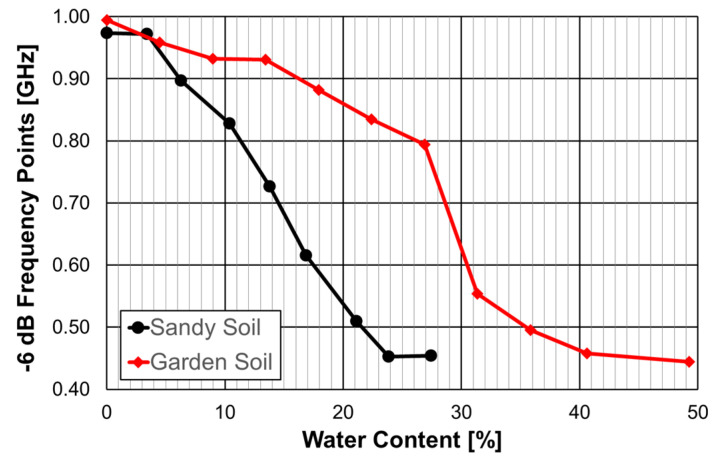
Experimental resonance frequency results for sandy and garden soils.

**Table 1 micromachines-15-00469-t001:** Physical characterization of Matryoshka geometries with two rings.

Configuration	L_1_ (mm)	L_2_ (mm)	L_c1_ (mm)
Config1	27.25	21.25	1.00
Config2	28.00	20.00	2.00
Config3	31.00	15.00	6.00
Config4	34.00	10.00	10.00

**Table 2 micromachines-15-00469-t002:** Main characteristics of open Matryoshka filter configurations.

Configuration	fr_1_ (GHz)		fr_2_ (GHz)		f_0_ (GHz)	BW * (%)
	Equation (2)	Simulation	Equation (2)	Simulation		
Config1	0.681	0.71	0.800	0.85	0.785	43.4
Config2	0.684	0.65	0.802	0.85	0.756	49.6
Config3	0.695	0.63	0.810	0.81	0.717	46.4
Config4	0.707	0.63	0.818	0.85	0.720	52.0

* Defined for a −10 dB reference level.

**Table 3 micromachines-15-00469-t003:** Physical characterization of open Matryoshka geometries with 2, 3, and 4 rings.

Configuration	N	L_1_ (mm)	L_2_ (mm)	L_3_ (mm)	L_4_ (mm)	L_c1_ (mm)	L_c2_ (mm)	L_c3_ (mm)	P_N_ (mm)
Config5	2	32.00	24.00	NA	NA	2.00	NA	NA	210.00
Config6	3			16.00	NA		2.00	NA	268.00
Config7	4				8.00			2.00	294.00

**Table 4 micromachines-15-00469-t004:** Main characteristics of open Matryoshka filter configurations with 2, 3 and 4 rings.

Configuration	N	P_N_ (mm)	fr_1_ (GHz)	fr_2_ (GHz)	f_0_ (GHz)	BW * (%)
Config5	2	210.0	0.55	0.69	0.627	48.7
Config6	3	268.0	0.43	0.53	0.501	48.4
Config7	4	294.0	0.37	0.51	0.462	50.3

* Defined for a −10 dB reference level.

**Table 5 micromachines-15-00469-t005:** Dimensions of the closed square Matryoshka geometry with two rings.

Configuration	L_1_	L_2_	L_c1_	w	g	P_2_ (mm)
Config1	22.0	12.0	3.5	1.5	1.0	129.0
Config2		7.0	6.0			114.0

**Table 6 micromachines-15-00469-t006:** Summary of the first resonance results for the square ring (SR) unit-cell FSSs with simple rings and closed (CM) and open (OM) Matryoshka geometries.

Unit-Cell Geometry	First Resonance Frequency (GHz)		Bandwidth * (%)	
	HPol	VPol	HPol	VPol
SR	2.56	2.56	35.4	35.4
CM	1.78	1.78	13.6	15.3
OM	1.78	1.01	10.9	8.1

* Defined for a −10 dB reference level.

**Table 7 micromachines-15-00469-t007:** Summary of the simulation results of the FSS with open Matryoshka unit-cells with 2 and 3 rings.

N	Area	f_r1_ (GHz)		Bandwidth * (%)		f_r2_ (GHz)		f_r3_ (GHz)	
	(mm^2^)	HPol	VPol	HPol	VPol	HPol	VPol	HPol	VPol
2	22 × 22	1.78	1.01	10.9	8.1	4.36	2.41	7.66	3.96
3		1.56	0.86	5.9	2.9	3.01	1.91	4.36	3.11

* Defined for a −10 dB reference level.

**Table 8 micromachines-15-00469-t008:** Radius of the circular Matryoshka unit-cells.

Configuration	Number of Rings	r_1_ (mm)	r_2_ (mm)	r_3_ (mm)	r_4_ (mm)	r_5_ (mm)
FSS1	1	9.0	NA			
FSS2	3		7.4	5.8	NA	
FSS3	5				4.2	2.6

**Table 9 micromachines-15-00469-t009:** Comparison of first resonance frequencies of the FSSs prototypes.

Configuration	Number of Rings	First Resonance Frequency (GHz)					
		Estimation	Simulation	Experimental			
				Θ = 0	Θ = 15°	Θ = 30°	Θ = 45°
FSS1	1	4.45 ^1^	4.10	4.224	4.211	4.133	4.120
FSS2	3	2.71 ^2^	2.70	2.846	2.833	2.833	2.768
FSS3	5	2.30 ^3^	2.20	2.378	2.352	2.404	2.417

^1^ Equation (10); ^2^ Equation (11); ^3^ Equation (12).

**Table 10 micromachines-15-00469-t010:** Physical characterization of open Matryoshka geometries with two, three, and four rings.

Configuration	N	L_1_ (mm)	L_2_ (mm)	L_3_ (mm)	L_c_ (mm)
Config1	2	28.0	20.0	NA	2.0
Config2			12.0		6.0
Config3			8.0		8.0
Config4	3	36.0	28.0	20.0	2.0
Config5		28.0	20.0	12.0	2.0

**Table 11 micromachines-15-00469-t011:** Main experimental characteristics of the five square filter configurations.

Configuration	fr_1_ (GHz)	fr_2_ (GHz)	f_0_ (GHz)	BW * (%)
Config1	0.700	0.805	0.769	45.4
Config2	0.770	0.980	0.876	49.0
Config3	0.840	1.120	0.964	51.4
Config4	0.375	0.420	0.421	43.8
Config5	0.540	0.660	0.626	46.4

* Defined for a −10 dB reference level.

**Table 12 micromachines-15-00469-t012:** Physical characterization of open Matryoshka circular geometries with two and three rings.

Configuration	N	R1 (mm)	R2 (mm)	R3 (mm)
Config1	2	14.0	12.0	NA
Config2			10.0	
Config3			8.0	
Config4			6.0	
Config5	3		12	10.0

**Table 13 micromachines-15-00469-t013:** Main experimental characteristics of the five circular filter configurations.

Configuration	fr_1_ (GHz)	fr_2_ (GHz)	f_0_ (GHz)	BW * (%)
Config1	0.961	1.091	1.039	35.7
Config2	0.941	1.101	1.047	43.1
Config3	0.981	1.181	1.099	45.2
Config4	1.031	1.311	1.172	48.0
Config5	0.701	0.811	0.790	36.7

* Defined for a −10 dB reference level.

**Table 14 micromachines-15-00469-t014:** Physical characterization of open Matryoshka geometry DGS configurations.

Configuration	L_1_ (mm)	L_2_ (mm)	L_c_ (mm)	w (mm)	g = s (mm)
Config1	17.0	11.0	1.5	1.5	1.0
Config2	15.5	9.5			
Config3	14.0	8.0			
Config4	12.5	6.5			

**Table 15 micromachines-15-00469-t015:** Comparison of first resonance characteristics of open Matryoshka and dumbbell DGS configurations.

Configuration	Matryoshka		Dumbbell	
	f_r1_ * (GHz)	BW (%)	f_r1Ma_/_fr1Db_ (%)	BW_Ma_/BW_Db_ (%)
Config1	2.07	28.8	52.8	19.0
Config2	2.39	29.3	55.1	20.0
Config3	2.88	29.8	59.1	20.0
Config4	3.72	29.6	67.9	22.3

* Defined for a −10 dB reference level.

**Table 16 micromachines-15-00469-t016:** Summary of the simulated radiation pattern results at the first resonance frequency.

Parameter		Matryoshka	SRR
Direction of maximum radiation (θ) (Degree)		≈180	≈0
Maximum gain (dBi)		4.9	4.6
Half-power beamwidth (Degree)	H-Plane	140	137
	E-Plane	87	86
FBR (dB)		−3.2	2.8

**Table 17 micromachines-15-00469-t017:** Summary of the main patch antenna characteristics.

Parameter		Without DGS	With Open Matryoshka DGS	With Closed Matryoshka DGS
First resonance frequency (GHz)		3.52	2.35	3.29
−10 dB bandwidth	(MHz)	92	53	108
	(%)	2.6	2.3	3.3
Gain (dBi)		4.24	2.40	3.55

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
