# Peer review of "Planar Printed Structures Based on Matryoshka Geometries: A Review"

_micromachines, 2024, doi:10.3390/mi15040469_

Round 1

Reviewer 1 Report

Comments and Suggestions for Authors

This article provides a systematic review of planar printed structures based on Matryoshka-like geometries, a unique and interesting area of research. The manuscript delves into how these structures leverage the advantages of planar printing technology and the characteristics of meandered nested Matryoshka geometries to achieve miniaturized, multi-resonance and/or wideband configurations. Some issues are as follows.

 1. The discussion on the physical behavior and design principles of these structures seems to need strengthening. It is suggested that the author provide more simulation and/or experimental results to support their theoretical analysis and the effectiveness of their designs.

 2. While some applications are mentioned, such as frequency-selective surfaces, filters, antennas, and sensors, there is a lack of specific application examples or case studies. Adding these contents would make the manuscript more comprehensive and help readers better understand the practical application potential of these structures.

 3. A detailed comparative analysis of these Matryoshka-like structures with existing technologies could highlight their advantages and potential limitations. This would provide readers with a more comprehensive perspective on the innovations of these structures relative to conventional structures.

 4. Although the article mentions future research and application prospects, these potential research areas could be discussed more in detail, especially pointing out the current technical bottlenecks and possible solutions for the future.

5. Actually, there are other interesting works, which represent an emerging field. (Appl Mech Rev 2023, 75, 020802, 2023; Compos Sci Technol 2024, 249, 110503; Frontiers in Materials 2021, 8,  603820; Engineering 2023, 28, 49-57). The authors are suggested to discuss this point in the manuscript.

Comments on the Quality of English Language

The author should improve the English expression to enhance the readability of the article.

Reviewer 2 Report

Comments and Suggestions for Authors

This paper presents a review of microstrip-based Matryoshka structures. The reviewer believes that integration of the following comments in the manuscript would improve their quality even further.

1. Please extend the introduction by describing the problem statement and available solutions.

2. Please highlight the importance of Matryoshka structures over other planar structures.

3. Please, discuss based on the applications or use cases.

4. Please include a table of comparison for each application and discuss in detail.

5. Please comment on the feasibility of Matryoshka structures for operation at high frequencies i.e. mmW and beyond.

Round 2

Reviewer 2 Report

Comments and Suggestions for Authors

The authors have addressed all of my comments from the previous revision round. I have no further concerns.